# A *Pseudomonas aeruginosa* TIR effector mediates immune evasion by targeting UBAP1 and TLR adaptors

Paul RC Imbert[1], Arthur Louche[1], Jean-Baptiste Luizet[1], Teddy Grandjean[2], Sarah Bigot[1], Thomas E Wood[3], Stéphanie Gagné[1], Amandine Blanco[1], Lydia Wunderley[4], Laurent Terradot[1], Philip Woodman[4], Steve Garvis[5], Alain Filloux[3], Benoit Guery[2] & Suzana P Salcedo[1,*]

## Abstract

Bacterial pathogens often subvert the innate immune system to establish a successful infection. The direct inhibition of downstream components of innate immune pathways is particularly well documented but how bacteria interfere with receptor proximal events is far less well understood. Here, we describe a Toll/interleukin 1 receptor (TIR) domain-containing protein (PumA) of the multi-drug resistant *Pseudomonas aeruginosa* PA7 strain. We found that PumA is essential for virulence and inhibits NF-κB, a property transferable to non-PumA strain PA14, suggesting no additional factors are needed for PumA function. The TIR domain is able to interact with the Toll-like receptor (TLR) adaptors TIRAP and MyD88, as well as the ubiquitin-associated protein 1 (UBAP1), a component of the endosomal-sorting complex required for transport I (ESCRT-I). These interactions are not spatially exclusive as we show UBAP1 can associate with MyD88, enhancing its plasma membrane localization. Combined targeting of UBAP1 and TLR adaptors by PumA impedes both cytokine and TLR receptor signalling, highlighting a novel strategy for innate immune evasion.

**Keywords** *Pseudomonas*; TIR domain; TLR adaptors; UBAP1; virulence
**Subject Categories** Microbiology, Virology & Host Pathogen Interaction
The EMBO Journal (2017) 36: 1869–1887

## Introduction

Microbial pathogen recognition by innate immune receptors initiates a progression of molecular interactions and signalling events assuring host defence. In bacterial infections, detection of surface components, such as peptidoglycan, lipopolysaccharides and flagellin by Toll-like receptors (TLR) 2, 4 and 5, respectively, is essential for induction of pro-inflammatory cytokines and type I interferon (IFN) responses. Specific sorting and signalling adaptor proteins bridge activated receptors with downstream kinases to initiate signalling cascades via Toll/interleukin 1 receptor (TIR) domains present on both the adaptors and the cytosolic face of TLRs (Brubaker *et al*, 2015). Upon TLR2 or TLR4 activation, the TIR-containing adaptor protein (TIRAP) recruits myeloid differentiation primary response 88 (MyD88) that interacts with the TLR via its TIR domain (Fitzgerald *et al*, 2001; Horng *et al*, 2001; Kagan & Medzhitov, 2006). MyD88 oligomerization and recruitment of specific kinases leads to the formation of myddosomes, signalling platforms that induce NF-κB translocation into the nucleus and subsequent transcription of pro-inflammatory associated genes (Nagpal *et al*, 2011; Bonham *et al*, 2014). TLR4 activation also results in induction of a type I IFN via another set of adaptors, TRAM and TRIF (Fitzgerald *et al*, 2001; Yamamoto *et al*, 2002; Kagan *et al*, 2008). In the case of the MyD88-dependent TLR5, the identity of a sorting adaptor remains undefined and the role of TIRAP unclear although it has been implicated in proper TLR5 signalling in epithelial cells (Choi *et al*, 2013).

Microbial pathogens have been shown to counter these host defence pathways. Most bacterial immune-modulatory proteins described to date rely on inhibition of downstream signalling components, such as MAP kinases and transcription factors (reviewed in Rosadini & Kagan, 2015). In contrast, few examples of direct blocking at the level of initial receptor–adaptor complexes are known. Some bacterial pathogens rely on TIR domain-containing proteins to perturb TIR-dependent interactions (Newman *et al*, 2006; Cirl *et al*, 2008; Salcedo *et al*, 2008, 2013), essential in innate immune signalling. The growing number of bacterial TIR proteins recently identified in both Gram-negative and Gram-positive human

1  Laboratory of Molecular Microbiology and Structural Biochemistry, Centre National de la Recherche Scientifique, University of Lyon, Lyon, France
2  EA 7366 Recherche Translationelle Relations Hôte-Pathogènes, Faculté de Médecine Pôle Recherche, Université Lille 2, Lille, France
3  MRC Centre for Molecular Bacteriology and Infection, Imperial College London, London, UK
4  School of Biological Sciences, Faculty of Biology Medicine and Health, University of Manchester, Manchester Academic Health Science Centre, Manchester, UK[†]
5  Laboratoire de Biologie et Modelisation, Ecole Normal Supérieur, UMR5239, Lyon, France
   *Corresponding author. Tel: +33 437652915; E-mail: suzana.salcedo@ibcp.fr
   [†]Correction added on 3 July 2017, after first online publication: affiliation 4 has been corrected

pathogens (Spear *et al*, 2012; Askarian *et al*, 2014; Zou *et al*, 2014) highlights the importance of this immune evasion strategy in disease. However, the molecular mechanisms underlying most TIR-dependent virulence strategies remain to be defined.

We focused on a previously uncharacterized TIR domain-containing protein of the multi-drug resistant pathogen *Pseudomonas aeruginosa* PA7, that we called PumA. *P. aeruginosa* PA7 lacks genes encoding the type III secretion system (T3SS) and its cognate effector proteins that are normally associated with strong induction of cell death, a hallmark of acute *P. aeruginosa* infections (reviewed by Filloux, 2011). In addition, PA7 does not show high lytic capacity towards epithelial cells due to exolysin A (ExlA) as described for the haemorrhagic pneumonia-causing strain of the same family (Elsen *et al*, 2014; Reboud *et al*, 2016). We thus took advantage of the absence of traditional virulence factors in this *P. aeruginosa* strain to study the molecular interactions involved in TIR-mediated bacterial targeting of events proximal to receptor–adaptor signalling complexes and to dissect PumA function. We found that the PumA *Pseudomonas* TIR domain-containing protein is essential for PA7 virulence conferring a previously unrecognized ability to *Pseudomonas* to down-modulate innate immune responses during infection. We show that PumA directly interacts with both TIRAP and MyD88 to control TLR signalling. Uniquely, it also targets the ubiquitin-associated protein 1 (UBAP1), a recently discovered component of the endosomal-sorting complex required for transport I (ESCRT-I; Stefani *et al*, 2011). UBAP1 is known to play a key role in selective sorting of ubiquitinated endosomal cargo on multi-vesicular bodies (MVB), via its interaction with VPS37A and other components of ESCRT-I namely TSG101 (Wunderley *et al*, 2014), as well as with the ESCRT regulator, His domain protein tyrosine phosphatase (HDPTP; Stefani *et al*, 2011). UBAP1 has been shown to control endosomal sorting of ubiquitinated EGFR (Stefani *et al*, 2011) as well as ubiquitin-dependent degradation of antiviral surface proteins (Agromayor *et al*, 2012) and integrins (Kharitidi *et al*, 2015). More recently, UBAP1 was shown to modulate steady-state trafficking of cytokine receptors in non-stimulated cells (Mamińska *et al*, 2016). UBAP1 is expressed in a wide range of tissues, but when deleted in mice, it is lethal for embryos (Agromayor *et al*, 2012).

We propose that this novel *Pseudomonas* effector modulates UBAP1 function, hence the name PumA (for *Pseudomonas* UBAP1 modulator A), which confers to this TIR domain-containing protein the distinctive ability to also interfere with cytokine receptor signalling. Targeting of both TLR adaptors and UBAP1 by PumA is not spatially restricted as we found UBAP1 can associate with MyD88 in host cells. Our results thus highlight a novel role of bacterial TIR domains and place UBAP1 sorting in the context of TLR signalling.

# Results

## PumA is required for *Pseudomonas aeruginosa* PA7 virulence *in vivo*

In *Pseudomonas,* TIR domain-containing proteins were first identified in an *in silico* study in *P. aeruginosa* and the plant pathogen *P. syringae* (Zhang *et al*, 2011). Analysis of currently available genomes shows that several plant strains encode such proteins as

well as additional human pathogenic strains of *P. stutzeri* and *P. aeruginosa*. The closest orthologue is found in the plant pathogen *P. viridiflava*. The TIR domain of PumA spans the first 136 amino acids of PumA (Appendix Fig S1A and B), with no significant sequence/structure homologies detected for the C-terminal domain (amino acid 137–303) and no signal peptide. Analysis of the PA7 genome shows *pumA* (*PSPA7_2375*) is within the genomic island RGP56, which displays a G+C content of 58.5% in contrast to the average 66.5% in the remaining genome. Interestingly, using Geneious (Kearse *et al*, 2012), we found the *pumA* gene itself has an even larger reduction in G+C content (46.6%) (Appendix Fig S1C), suggesting that it is not a conserved gene within its immediate genetic context.

We assessed the potential role of PumA in virulence by infecting the nematode *Caenorhabditis elegans*, a well-established model for *P. aeruginosa* allowing for rapid assessment of virulence (Garvis *et al*, 2009). Infection with the highly virulent strain *P. aeruginosa* PA14 which contains virulence factors such as the T3SS but no TIR protein resulted in 50% lethality at day 5. The PA7 wild-type strain caused 50% lethality 7 days after inoculation. In contrast, we found that the PA7 Δ*pumA* mutant showed a slight but significant attenuation in virulence in *C. elegans* (Fig 1A). These differences were not due to an *in vitro* growth defect of the mutant (Appendix Fig S2A) nor to a problem in expression of PumA in the wild-type *P. aeruginosa* PA7 strain (Appendix Fig S2B).

We then used an acute *in vivo* infection model to evaluate the involvement of *pumA* in *P. aeruginosa* induced lung injury. Mice infected with Δ*pumA* showed a clear increased survival compared to wild-type strain (Fig 1B). A dose of $4.10^7$ CFU of PA7 induced 100% lethality after 52 h against 62.5% survival after 96 h for the Δ*pumA* mutant. Bacterial clearance and cellular recruitment were then analysed with a lower inoculum of $3.10^7$ CFU. PA7Δ*pumA* infected mice showed decreased cell recruitment (Fig 1C) and an enhanced lung bacterial clearance in bronchoalveolar lavages (BAL) compared to the wild-type strain (Fig 1D). The bacterial dissemination measured with the spleen bacterial load was equivalent between the two groups (Fig 1E). Together these results show that PumA is required for *P. aeruginosa* PA7 infection.

## PumA inhibits NF-κB translocation into the nucleus during infection *in vitro*

As bacterial TIR proteins down-modulate NF-κB activation (Newman *et al*, 2006; Cirl *et al*, 2008; Salcedo *et al*, 2008, 2013; Spear *et al*, 2012; Askarian *et al*, 2014; Zou *et al*, 2014), we infected the lung carcinoma epithelial cell line A549, a well-established cellular model for *Pseudomonas* infection and analysed NF-κB translocation into the nucleus after one hour of infection by confocal microscopy. We developed an automated analysis of p65/RelA fluorescence in relation to DAPI labelling using a specific ImageJ plugin from images obtained by confocal microscopy (Fig EV1A) which allowed us to clearly differentiate between TNFα-treated and mock-infected cells (Figs 2A and EV1B). Infection with the three heat-killed *P. aeruginosa* strains, wild-type PA7, isogenic mutant Δ*pumA* or wild-type PA14 resulted in significant induction of NF-κB translocation into the nucleus, although to a lower level than TNFα-treated cells (Fig 2A). When cells were infected with PA7, there was no significant induction of

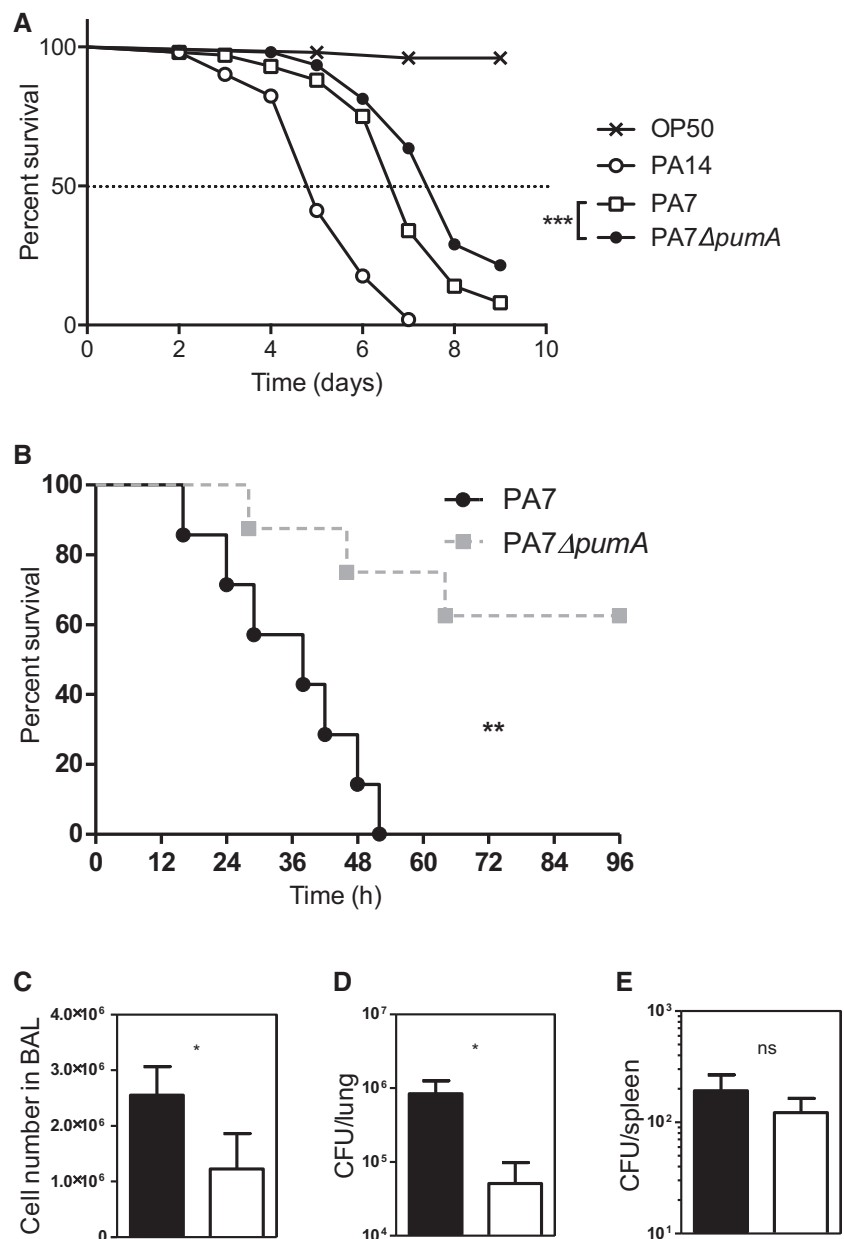

**Figure 1.  PumA is required for *Pseudomonas aeruginosa* PA7 virulence *in vivo*.**

A    *Caenorhabditis elegans* survival curve. Fifty *C. elegans* were infected with *E. coli* OP50 and with highly virulent strain *P. aeruginosa* PA14. One hundred *C. elegans* were infected with *P. aeruginosa* PA7 and PA7 Δ*pumA*. Test of Mantel–Cox was used with ***P = 0.0002.

B    To establish an *in vivo* model of acute infection, mice were intranasally infected with $4 \times 10^7$ CFU *P. aeruginosa* PA7 or PA7Δ*pumA* strains (*n* = 7/group). Lethality was monitored for 96 h, and a test of Mantel–Cox was used, with **P = 0.0035.

C–E  Mice were intranasally infected with $3 \times 10^7$ CFU *P. aeruginosa* PA7 or PA7Δ*pumA* strains (*n* = 7/group). Cells from bronchoalveolar lavage (BAL) were counted (C). Bacterial load in the lungs (D) and dissemination (E) were assessed through cultured lung or spleen homogenate. Non-parametric two-tailed Mann–Whitney test was carried out with (C) *P = 0.0173, (D) *P = 00364 and (E) P = 0.3629. All data correspond to mean ± standard error.

NF-κB when compared with the mock-infected negative control (Fig 2B), suggesting PA7 blocks NF-κB translocation into the nucleus. In contrast, Δ*pumA* infection promoted NF-κB nuclear translocation, attaining activation levels similar to those observed with heat-killed bacteria. The inability of *pumA* mutants to block NF-κB nuclear transport was complemented by chromosomal expression of this gene under an arabinose-inducible promoter (Fig 2C). Importantly, when *pumA* expression was repressed with glucose, no complementation of NF-κB inhibition was observed (Fig 2B). Addition of arabinose had no effect on NF-κB

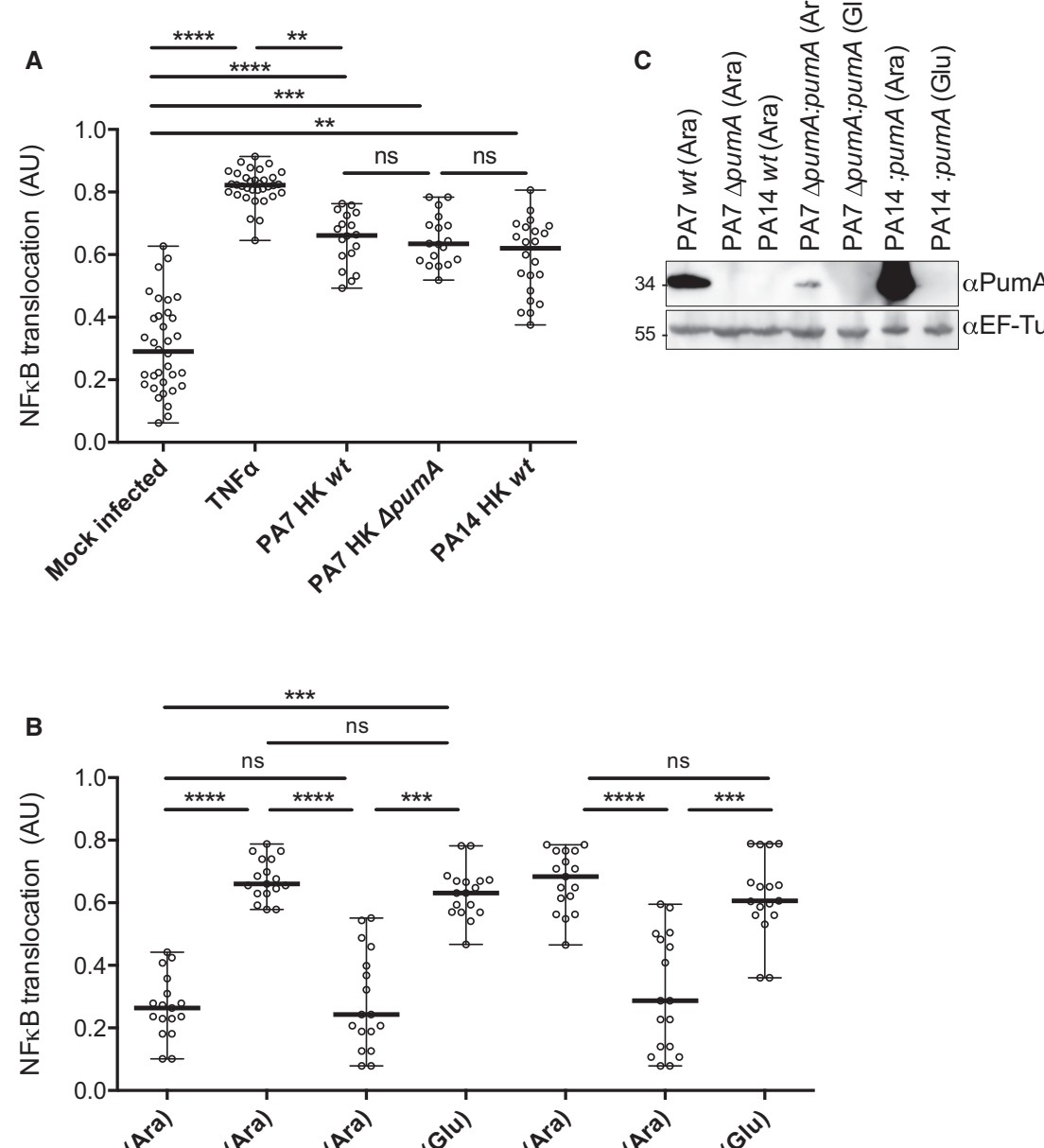

**Figure 2. PumA is essential for control of NF-κB translocation into the nucleus during *Pseudomonas aeruginosa* infection of human A549 lung epithelial cells.**

A   Quantification of fluorescence ratio between nuclear NF-κB (p65) and nuclear DAPI staining. Negative control corresponds to uninfected cells that underwent all steps of the experiment. Positive control corresponds to full NF-κB activation with TNFα (1 μg/ml). A549 cells were incubated for 1 h with heat-killed (HK) bacteria to establish maximum activation induced by *Pseudomonas* infection.

B   Cells were infected for 1 h with either *P. aeruginosa* PA7 wt, ΔpumA, ΔpumA:pumA (Ara) induced with 1% arabinose, ΔpumA:pumA (Glu) repressed with 0.5% glucose, PA14 wt, PA14:pumA (Ara) induced with 1% arabinose and PA14:pumA (Glu) repressed with 0.5% glucose. For consistency, arabinose was also included for the infections with wild-type and deletion mutant strains.

C   Western blots from representative inocula used for the infection experiments, showing expression of PumA (34 kDa) in the different *P. aeruginosa* strains visualized using a polyclonal rabbit anti-PumA with control blot against the standard cytoplasmic protein EF-Tu (45 kDa) below.

Data information: For (A) and (B), between 200 and 400 cells per condition were counted and data correspond to median ± standard error from three independent experiments. Non-parametric one-way ANOVA Kruskal–Wallis test, with Dunn's multiple comparisons test was performed; ****P < 0.0001, ***P < 0.001, **P < 0.01.

translocation (Fig 2B versus EV1B). Furthermore, absence of NF-κB nuclear translocation was not due to reduced immune detection of the mutant strain as incubation of host cells with heat-killed Δ*pumA* resulted in equivalent levels of NF-κB activation to the wild-type PA7 (Fig 2A). To further support that the differences observed relate to PumA and are not indirect, we verified that all strains showed equivalent levels of membrane permeability (Fig EV2A), the global protein composition of the cell envelope was not altered in a *pumA* mutant (Fig EV2B), and no differences were observed in cytotoxicity (Fig EV2C and D) nor host cell adhesion (Fig EV2E) between wild-type PA7 and isogenic Δ*pumA* mutant. Together these results show that PumA is responsible for *P. aeruginosa* PA7 inhibition of NF-κB nuclear translocation during infection.

We then investigated whether expression of PumA alone could confer the ability to block NF-κB translocation to a different *Pseudomonas* strain. We chose PA14, which does not contain *pumA* and is known to be more virulent due to the presence of several virulence factors, namely those secreted by the T3SS. As expected, cells infected with wild-type PA14 showed high levels of NF-κB translocation into the nucleus (Fig 2B). Induction of *pumA* from the PA14 chromosome, which did not impact membrane permeability (Fig EV2A), was sufficient to enable this highly virulent strain to block NF-κB accumulation in the nucleus of infected cells (Fig 2B). These data indicate that PumA expression in *P. aeruginosa* is necessary and sufficient for NF-κB inhibition, highlighting its central role in immune evasion.

### PumA translocation into host cells during infection *in vitro*

We next sought to determine whether PumA could be secreted by *Pseudomonas*. Fractionation of bacterial cells grown in liquid culture indicates that PumA is mostly cytoplasmic and to a lesser extent associated with the inner membrane (Appendix Fig S3A). The protein was not detected in the outer membrane fractions nor could it be found in the supernatant indicating absence of secretion into the extracellular milieu *in vitro*. To determine whether PumA could be found inside host cells during infection, we fused chromosomal PumA with the TEM1 β-lactamase. Although the presence of other β-lactamases in PA7 and/or potential bacterial lysis resulted in non-specific cleavage of the CCF2 substrate within host cells infected with the wild-type strain, significant levels of coumarin fluorescing cells following infection with a strain containing PumA-TEM1 (Appendix Fig S3B) suggest that PumA is translocated into host cells during infection.

### PumA is associated with both TIRAP and MyD88 at the plasma membrane and intracellular compartments

To determine the mechanism by which PumA interferes with NF-κB activity, PumA was expressed in mammalian cells. We found PumA localized mostly at the plasma membrane, with some intracellular distribution, independently of the tag and in both immortal HeLa cells (Fig EV3A, top panel) and primary mouse embryonic fibroblasts (MEFs, Fig EV3B). As this localization was reminiscent of that of the TLR adaptor TIRAP (Fig EV3A and B), we co-transfected cells with PumA and TIRAP. We found extensive co-localization, in particular at the plasma membrane in both HeLa and MEFs (Fig 3A

and B). We observed these results with any combination of tags (HA, Myc or GFP) for both proteins.

In contrast with TIRAP, MyD88 is mostly localized in intracellular structures that do not label the plasma membrane. We therefore co-expressed MyD88 with PumA. Surprisingly, we found enrichment of PumA in a proportion of MyD88-positive structures in both cell types although to a lesser extent than that observed with TIRAP (Fig 3A and B). These results were confirmed by structured illumination microscopy (SIM) to enable imaging at higher resolution (Fig 3C and D). PumA enrichment was observed with PumA tagged with Myc or HA and MyD88 fused to either HA, FLAG or Myc. Curiously, this phenotype was exacerbated when GFP-PumA, normally at the cell surface (Fig EV3) was co-expressed with MyD88, resulting in the majority of GFP-PumA being recruited to MyD88-positive compartments (Fig EV3C).

### The TIR domain of PumA is responsible for interaction with both TIRAP and MyD88

We then investigated whether the TIR domain present in the first 136 amino acids of PumA was responsible for membrane targeting. PumA$_{1-136}$ was also efficiently targeted to the plasma membrane (Fig 4A). However, unlike TIRAP and another bacterial TIR protein BtpA/TcpB which are known to interact with specific phospholipids of the plasma membrane (Kagan & Medzhitov, 2006; Radhakrishnan *et al*, 2009), PumA and PumA$_{1-136}$ did not show any lipid binding properties when incubated with phosphoinositide phosphate strips (Fig 4B). We then tested whether PumA could interact with TIRAP, which could explain its membrane localization. We found that TIRAP-GFP and Myc-PumA co-immunoprecipitated (co-IP) suggesting TIRAP and PumA could be part of the same complex (Fig 4C). This association was confirmed using purified His-tagged PumA or PumA$_{1-136}$ immobilized on Ni-NTA resin, which both retained HA-TIRAP (Fig 4D).

As we had also observed enrichment of PumA in MyD88-positive compartments, we investigated whether PumA could interact with this adaptor protein. Although we did not observe and interaction between GFP-MyD88 and Myc-PumA (Fig 4C) nor HA-PumA and Myc-MyD88 (Appendix Fig S4A), using co-IP assays, His-PumA or PumA$_{1-136}$ were able to retain HA-Myd88 (Fig 5A), suggesting PumA is also able to interact with MyD88. To confirm these results, we took advantage of the strong enrichment in MyD88-positive compartments when Myd88 is co-expressed with the GFP-tagged version of PumA (Fig EV3C) and carried out co-IP in these conditions. Indeed, GFP-PumA and Myc-Myd88 could be co-immunoprecipitated as well as GFP-PumA and Myc-TIRAP (Fig 5B), suggesting that PumA can interact with MyD88. As a control for non-specific TIR–TIR interactions, we tested the ability of PumA to interact with TLR2, also by co-IP. Indeed, GFP-PumA could interact with FLAG-TIRAP but not FLAG-TLR2, suggesting some level of specificity in PumA targeting (Appendix Fig S4B and C).

Finally, we co-expressed in *E. coli* His-PumA$_{1-136}$ with His-MBP (Fig 5C), His-MBP-TIRAP (Fig 5D) or His-MBP-MyD88 (Fig 5E) and we could clearly see co-elution of both TIRAP and MyD88 in contrast to the His-MBP control (also see Appendix Fig S4D).

We next sought to determine whether the C-terminus of PumA$_{137-303}$ could also participate in these interactions. Lack of expression of His-PumA$_{137-303}$ in *E. coli* prevented us from purifying

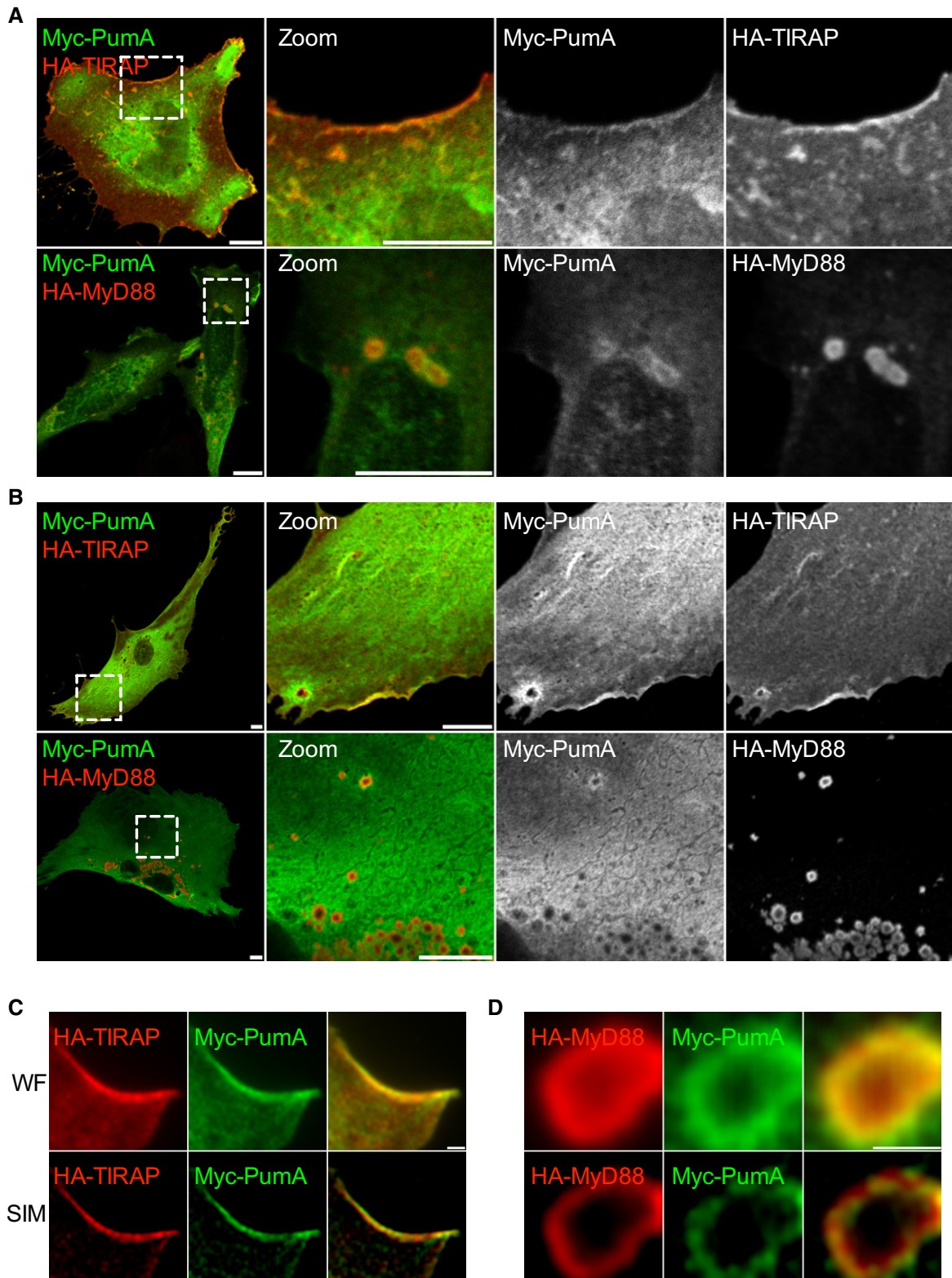

**Figure 3.    PumA co-localizes with TIRAP at the plasma membrane and to a lesser extent with intracellular MyD88, when ectopically expressed in host cells.**

A, B    Confocal microscopy of HeLa cells (A) and mouse embryonic fibroblast (MEFs) (B) co-expressing Myc-PumA and adaptor proteins HA-TIRAP (top panel) and HA-MyD88 (bottom panel). Cells were fixed after 10 h of transfection. Scale bars correspond to 10 μm.

C, D    Representative micrographs obtained by super resolution structure illumination microscopy (SIM) of MEFs co-expressing (C) Myc-PumA and TIRAP and (D) Myc-PumA and HA-MyD88. Wide field (WF) is shown in top panels and structured illumination of wide field (SIM) in bottom panels. Scale bars correspond to 1 μm.

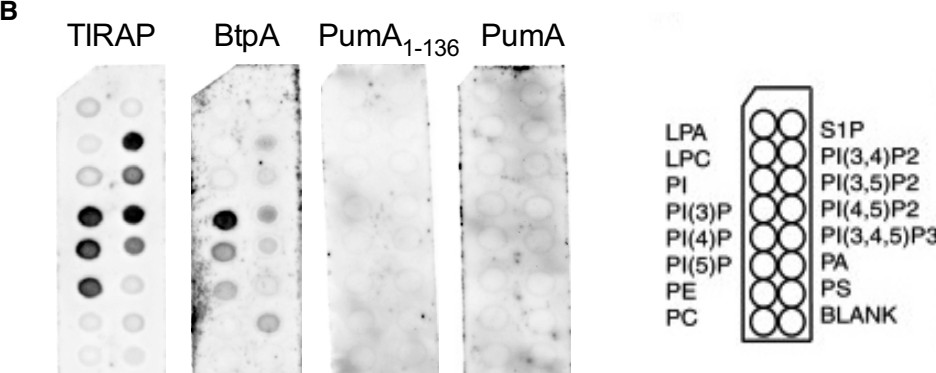

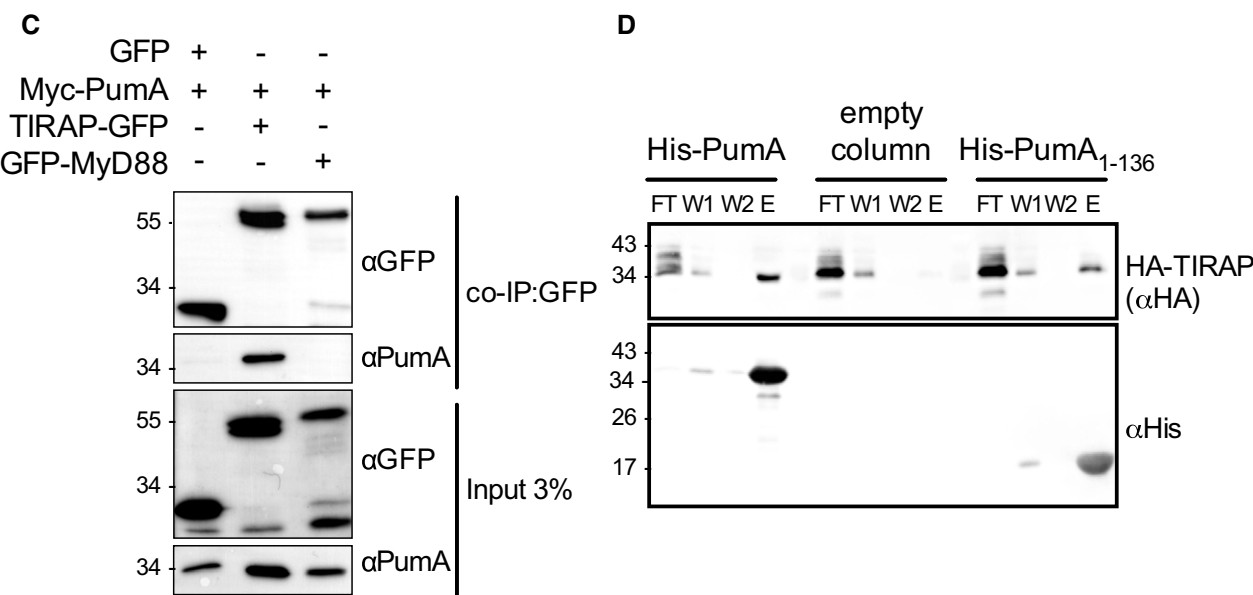

**Figure 4.  The TIR domain of PumA is required for plasma membrane targeting and interaction with TIRAP.**

A   Confocal microscopy of HeLa cells co-expressing HA-PumA$_{1-136}$ (TIR domain) and Myc-TIRAP labelled with anti-HA (green) and anti-Myc (red) antibodies. Cells were fixed after 10 h of transfection. Scale bars correspond to 10 μm.

B   PIP strip binding of purified TIRAP, BtpA (*Brucella* TIR protein A), PumA$_{1-136}$ or full-length PumA. Scheme on the right indicates the identity of each phospholipid on the PIP strips.

C   Co-immunoprecipitation (co-IP) assay from cells co-expressing Myc-PumA and GFP (control) or Myc-PumA with either TIRAP-GFP or GFP-MyD88. The co-IP was revealed using an anti-PumA antibody, binding to beads using an anti-GFP antibody and the inputs (shown below) using both anti-GFP and anti-PumA antibodies.

D   Pull-down assay using extracts from cells expressing HA-TIRAP against His-PumA or His-PumA$_{1-136}$ immobilized on a Ni-NTA resin. Empty column was used as a control for non-specific binding. Interactions were visualized by Western blotting using anti-HA antibody, and column binding with anti-His (lower blot). Flow-through (FT), two washes (W1 and W2) and elution (E) are shown for each sample.

Source data are available online for this figure.

    

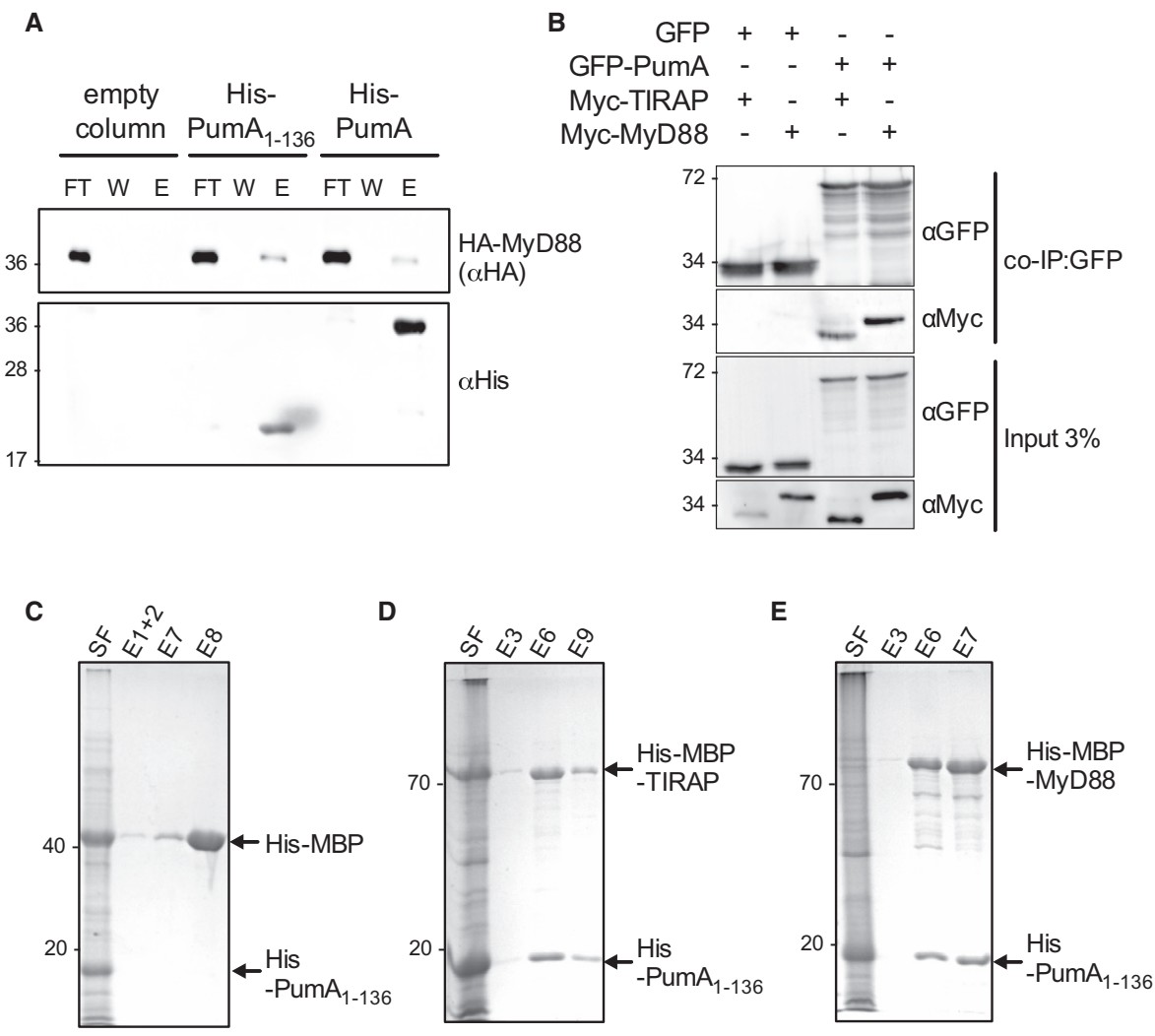

**Figure 5. PumA is also capable of interacting with MyD88.**

A Pull-down assay using extracts from cells expressing HA-MyD88 against His-PumA or His-PumA$_{1-136}$ immobilized on a Ni-NTA resin. Empty column was used as a control for non-specific binding. Interactions were visualized by Western blotting using anti-HA antibody, and column binding with anti-His (lower blot). Non-bound fraction (FT), last wash (W) and elution (E) are shown for each sample and the molecular weights indicated (kDa).

B Co-immunoprecipitation (co-IP) assay from cells expressing GFP-PumA and either Myc-TIRAP or Myc-MyD88. GFP was used as a control for non-specific binding. The co-IP was revealed using an anti-Myc antibody, the fraction bound to GFP-trapping beads using an anti-GFP antibody and the inputs (shown on the bottom two images) using both anti-Myc and anti-GFP antibodies.

C–E Co-purification of His-PumA$_{1-136}$ co-expressed in *E. coli* BL21 with either (C) His-MBP (control), (D) His-MBP-TIRAP or (E) His-MBP-MyD88. Interactions were visualized with coomassie blue stained gels. Soluble fraction (SF) and selected elutions (E) are shown for each sample and the molecular weights indicated (kDa).

Source data are available online for this figure.

this domain. We therefore carried out co-IP experiments with GFP-PumA$_{137-303}$ expressed in host cells. We could not detect any interaction between PumA$_{137-303}$ and TIRAP (Fig EV4A) nor between PumA$_{137-303}$ and MyD88 (Fig EV4B). However, it is important to note that expression of PumA$_{137-303}$ results in loss of plasma membrane localization. Instead, we observed formation of cellular aggregates that are positive for FK2 labelling (Fig EV4C), which recognizes mono- and poly-ubiquitinated proteins and could correspond to misfolded protein. For this reason, we cannot completely exclude a role of the C-terminus of PumA in these interactions. Nonetheless, our data identify TIRAP and MyD88 as host cell targets

of PumA, which mediates interaction with these adaptor proteins via its TIR domain.

## PumA interacts with the ESCRT-I component UBAP1

As PumA was able to interact with both TIRAP and MyD88, two key adaptors in TLR signalling, we hypothesized PumA's function was to block all immune pathways dependent on these adaptors. Using an *in vitro* luciferase assay, we tested key immune receptors implicated in *Pseudomonas* infection. Surprisingly, we found PumA could not only block TLR4, TLR5 and IL-1β but also the TNF receptor,

which is not dependent on TIR–TIR interactions (Fig EV5A). TNFR1 inhibition was specific to PumA as expression of the *Brucella* TIR domain-containing protein BtpA/TcpB did not have any significant effect (Fig EV5B). We therefore carried a yeast two-hybrid screen to identify an alternative target of PumA and found UBAP1, a key component of the ESCRT-I mediating trafficking and sorting of ubiquitinated cargo proteins on MVBs (Stefani *et al*, 2011; Agromayor *et al*, 2012). This interaction was confirmed by co-IP from cells co-expressing PumA and UBAP1 (Fig 6A and B) as well as by pull-down using purified PumA or PumA$_{1-136}$ and cell extracts with either streptavidin-tagged UBAP1 (Fig 6C) or Myc-UBAP1 (Appendix Fig S5A). These results were specific to PumA as the *Brucella* TIR protein BtpA/TcpB did not show any interaction (Fig 6C). Furthermore, PumA$_{137-303}$ could not co-IP UBAP1 (Fig EV4D) supporting a role of the TIR domain in targeting UBAP1. Not surprisingly, microscopy analysis of cells expressing both PumA and UBAP1 showed significant co-localization at the plasma membrane and intracellular compartments (Appendix Fig S5C).

We next investigated whether PumA is interacting with UBAP1 in the context of the ESCRT-I machinery. As over-expression of UBAP1 could result in its mislocalization, we carried out endogenous co-IP from cells expressing HA-PumA. Full-length PumA not only interacted very efficiently with endogenous UBAP1 but more importantly also co-immunoprecipitated TSG101 (Fig 6D), confirming PumA is targeting the ESCRT-I machinery. As expected, PumA also interacted with endogenous TIRAP (Fig 6D). The TIR domain of PumA only weakly interacted with endogenous UBAP1 and TIRAP (Fig 6E), whereas the C-terminus of PumA showed no interactions (Fig 6F).

While we were conducting this work, another study reported UBAP1 participates in control of TNFR1 and other cytokine receptor trafficking (Mamińska *et al*, 2016). Our data along with this recently published study thus suggest that PumA interaction with UBAP1 results in inhibition of the TNF receptor-mediated pathway. To determine whether PumA was targeting two different types of cellular compartments, one with UBAP1 controlling the TNFR pathway and another containing TLR adaptors, we analysed whether UBAP1 was excluded from TIRAP and MyD88 containing compartments. We first analysed their intracellular localization following transfection as we were not able to detect endogenous UBAP1 with currently available antibodies. Extensive co-localization was observed at the plasma membrane and intracellular structures when co-expressing UBAP1 and TIRAP (Fig 7A), with no visible impact on the normal distribution of TIRAP. However, in the case of MyD88, co-expression with UBAP1 resulted in accumulation of this adaptor at the plasma membrane, not seen in cells expressing MyD88 alone (Fig 7A, bottom panel and B). Quantification of membrane enrichment of MyD88 in cells expressing UBAP1 showed MyD88 membrane association was even more striking in the presence of UBAP1 (Fig 7C) than that observed when co-expressing TIRAP with MyD88 (Kagan & Medzhitov, 2006), suggesting UBAP1 could be participating in MyD88 intracellular sorting.

To determine whether UBAP1 could be interacting with these TLR adaptors, we carried out biochemical analysis of cells co-expressing UBAP1 and either TIRAP or MyD88. We could efficiently detect an interaction between UBAP1 and MyD88 by co-IP (Appendix Fig S5D and E) but not UBAP1 and membrin (Appendix Fig S5E), used as a control eukaryotic protein with the same tag. In the case of TIRAP, the co-IP was much less efficient

(Appendix Fig S5D). These results suggest that UBAP1 may be associated with MyD88-containing compartments and to a lesser extent TIRAP, consistent with our microscopy observations. To confirm these results and ensure these interactions were taking place with UBAP1 in the context of the ESCRT-I, we determined whether MyD88 and TIRAP could interact with endogenous UBAP1 and TSG101. We found that HA-MyD88 co-immunoprecipitated both components of the ESCRT-I as well as endogenous TIRAP (Fig 7D), as expected. However, we did not observe an interaction between HA-TIRAP and endogenous UBAP1 nor TSG101 (Fig 7E), suggesting that only MyD88 can be found associated with the ESCRT-I.

Overall, these data suggest that PumA mediates interactions with UBAP1 in the context of ESCRT-I, which can itself associate with the TLR adaptor MyD88, also targeted by this *P. aeruginosa* effector protein.

### *Pseudomonas aeruginosa* PA7 induces a decrease of TNFR1 in a PumA-dependent manner during infection *in vitro*

It is well described in the literature that inhibition of UBAP1 induces intracellular accumulation of EGFR, LTβR and TNFR1 (Stefani *et al*, 2011; Mamińska *et al*, 2016). To establish a link between PumA interaction with UBAP1 and the ability of PumA to reduce TNFα-dependent signalling (Fig EV5A), we analysed the levels of TNFR1 during infection. In wild-type PA7 infected A549 cells, we observed a decrease of TNFR1 compared to the negative control (Fig 7F). In contrast, the mutant lacking PumA was not able to reduce the levels of TNFR1 in infected cells and this phenotype could be fully restored by expression of PumA in the complemented strain. This is consistent with PumA targeting of UBAP1 and enhancing its activity during infection *in vitro*. Interestingly, we did not see any impact on the overall levels of TIRAP during infection (Fig 7F) suggesting that PumA is not inducing TIRAP degradation as was previously reported for BtpA (Sengupta *et al*, 2010).

## Discussion

Many pathogens have developed sophisticated strategies to evade or modify host immune responses to their advantage. We have found that the TIR domain-containing protein PumA plays a major role in the virulence of multi-drug resistant *P. aeruginosa* PA7 strain. PumA ensures efficient inhibition of innate immune responses by interacting with MyD88 and TIRAP, key adaptor proteins for IL-1R and the main relevant TLRs in *Pseudomonas* infection (TLR4 and TLR5), as well as UBAP1 which regulates cytokine receptor pathways. These results identify UBAP1 as a novel cellular target for bacterial pathogens.

*Pseudomonas aeruginosa* is an important human pathogen associated with high level of mortality in nosocomial infections and cystic fibrosis patients. Most *P. aeruginosa* strains rely on a multitude of virulence factors to control host cellular pathways, including effectors delivered by the T3SS. However, in a cystic fibrosis context, colonizing strains modulate levels of expression of some of these virulence factors (Hauser *et al*, 2011), namely downregulation of the T3SS (Jain *et al*, 2004) and undergo a remarkable accumulation of pathoadaptive mutations (Marvig *et al*, 2014). The

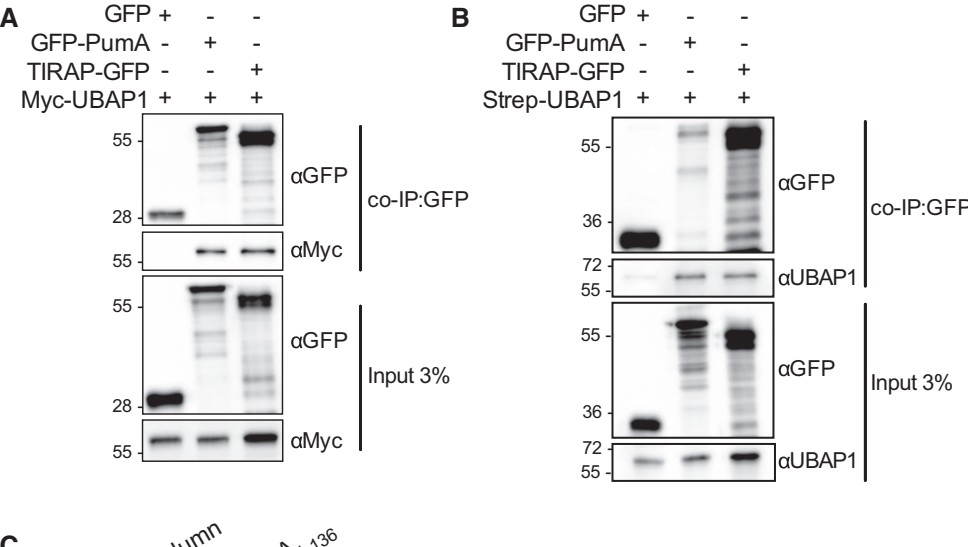

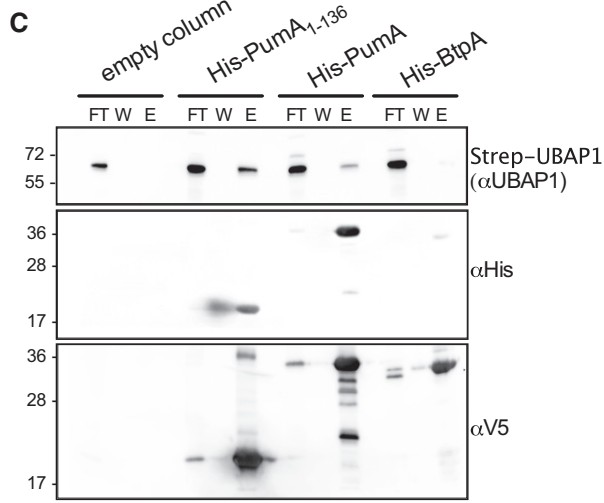

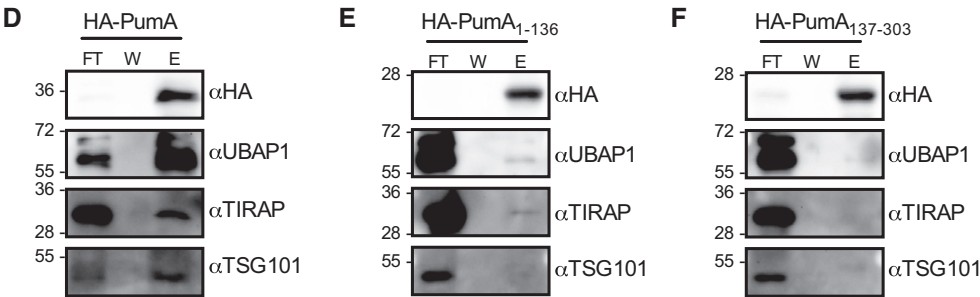

**Figure 6.  Identification of UBAP1 as a novel host protein targeted by the bacterial TIR domain of PumA.**

A, B   Co-immunoprecipitation (co-IP) assay from cells expressing Myc-UBAP1 (A) or Strep-UBAP1 (B) with either GFP, GFP-PumA or TIRAP-GFP. The co-IPs were revealed using an anti-Myc (A) or anti-UBAP1 (B) antibodies, the fractions bound to GFP-trapping beads using an anti-GFP antibody and the inputs using anti-Myc, anti-GFP or anti-UBPA1 antibodies as indicated.

C   Pull-down assay using extracts from cells expressing Strep-UBAP1 against His-PumA or His-PumA$_{1-136}$ immobilized on a Ni-NTA resin. Empty column was used as a control for non-specific binding. Interactions were visualized by Western blotting using anti-UBAP1 antibody, and column binding with anti-His (middle blot), followed by anti-V5 (lower blot), necessary for detection of BtpA, which for reasons we do not understand cannot be easily detected with the anti-His antibody (Appendix Fig S5B).

D–F   Endogenous co-IP from cells expressing (D) HA-PumA, (E) HA-PumA$_{1-136}$ and (F) HA-PumA$_{137-303}$. The fractions bound to HA-trapping beads were probed with anti-HA, anti-UBAP1, anti-TIRAP and anti-TSG101 antibodies. Non-bound fraction (FT), last wash (W) and elution (E) are shown for each sample and the molecular weights indicated (kDa).

Source data are available online for this figure.

    

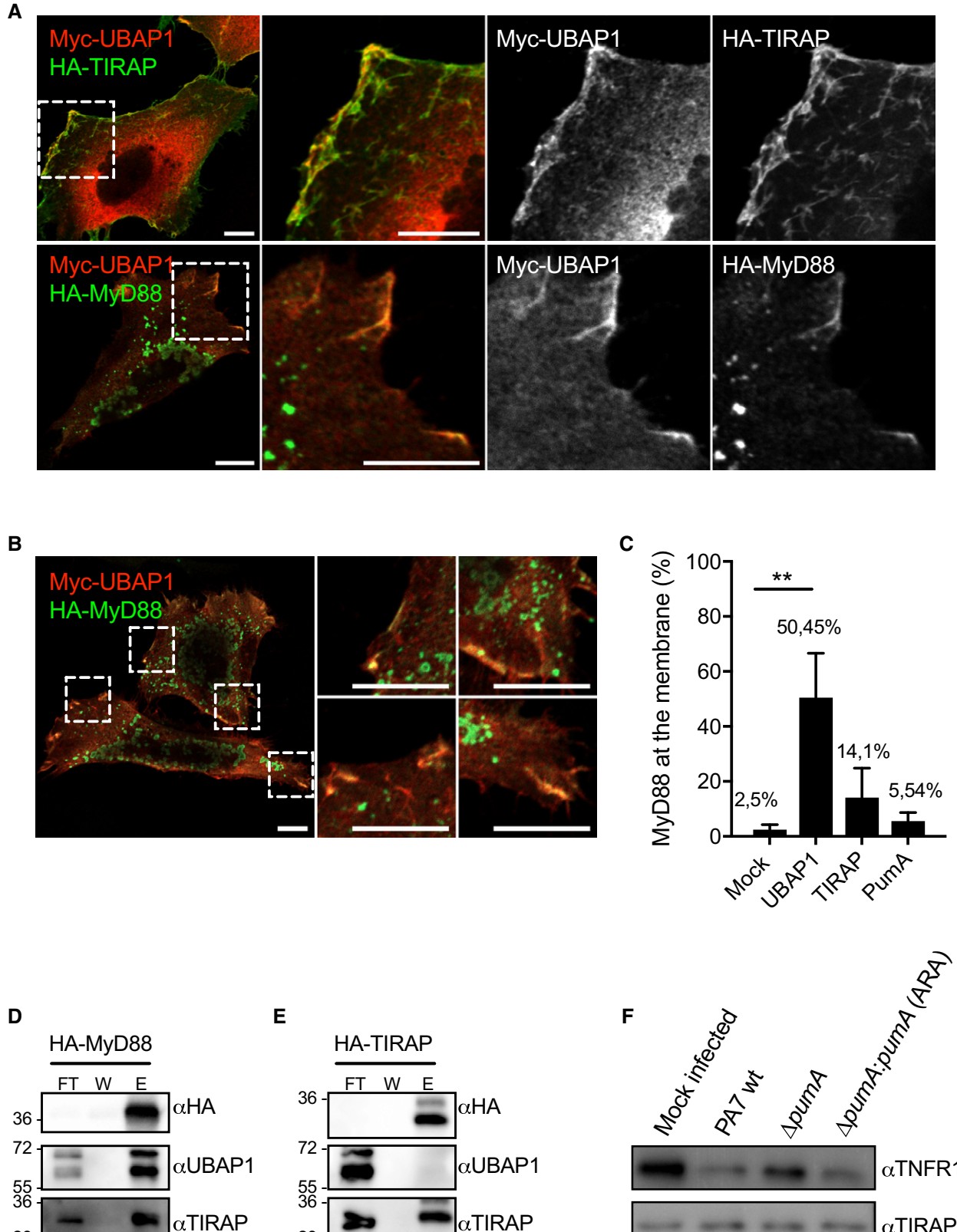

**Figure 7.**

◄

**Figure 7.  Analysis of the impact of UBAP1 on TIRAP and MyD88.**

A      Representative micrographs obtained by confocal microscopy of HeLa cells co-expressing Myc-UBAP1 (red) and adaptor proteins HA-TIRAP (green, top panel) or HA-MyD88 (green, bottom panel). Cells were fixed after 10 h of transfection. Scale bars correspond to 10 µm.

B      Different zoomed images showing HA-MyD88 (green) recruitment to the plasma membrane in the presence of Myc-UBAP1 (red). Scale bars correspond to 10 µm.

C      Quantification of plasma membrane localization of MyD88 in cells expressing MyD88 alone or with either UBAP1, TIRAP or PumA. At least 200 cells were enumerated in three independent experiments, and membrane localization was defined under the strict criteria of clear line at the plasma membrane. Cells with MyD88-positive vesicles in close proximity to the plasma membrane were not counted as positive. Non-parametric one-way ANOVA Kruskal–Wallis test was performed, with Dunn's multiple comparisons test.  **$P < 0.01$.

D, E   Endogenous co-IP from cells expressing (D) HA-MyD88 and (E) HA-TIRAP. The fractions bound to HA-trapping beads were probed with anti-HA, anti-UBAP1, anti-TIRAP and anti-TSG101 antibodies. Non-bound fraction (FT), last wash (W) and elution (E) are shown for each sample and the molecular weights indicated (kDa).

F      Western blot of TNFR1 in A549 cells infected for 1 h with either *P. aeruginosa* PA7 wt, Δ*pumA* or Δ*pumA:pumA* (Ara) induced with 1% arabinose. A mock-infected sample was included as a negative control. The same blot was also probed for TIRAP and actin to control loading.

Source data are available online for this figure.

PA7-related *P. aeruginosa* strains lack the 20-Kb-long genomic region encoding the T3SS core components and all genes encoding secreted effectors but contains several additional genomic islands and potential novel virulence factors (Pirnay *et al*, 2009; Roy *et al*, 2010; Cadoret *et al*, 2014; Freschi *et al*, 2015). In some of these strains, such as CLJ1, an exolysin secreted by a two-partner secretion system is responsible for hypervirulence (Elsen *et al*, 2014). However, in PA7, this exolysin is detected at only low levels in the secretome and is not responsible for cytotoxicity (Reboud *et al*, 2016), suggesting an alternative pathogenicity mechanism. In this context, we hypothesize that PumA might be underlying an alternative pathogenicity mechanism to allow PA7 persistence within a host. Consistently, we observed a clear attenuation in virulence for a PA7 strain lacking *pumA* in both *C. elegans* and in a mouse lung infection model. Interestingly, no impact in the ability of the *pumA* mutant to disseminate systemically was observed suggesting a role in control of local pathology. This type of *P. aeruginosa* infection based on persistence and colonization rather than rapid cytotoxicity could be relevant in specific clinical contexts such as infection of wound and burn patients, aggravated by the high level of multi-drug resistance. It is interesting to note that other *Pseudomonas* contain a TIR domain protein, namely several strains pathogenic in plants. In this context, it will be interesting to analyse the role of the ortholo- gous TIR protein in the plant pathogens *P. syringae* or *P. viridiflava* with over 90% identity to PumA in amino acid sequence for the TIR domain, regarding control of plant responses as these functions may be relevant across taxonomic kingdoms.

Pseudomonas* is not the only bacterial pathogen to take advan- tage of the TIR domain to engage TIR–TIR interactions which are essential components of innate immune signalling. Bacterial target- ing of TLRs has been best described for uropathogenic *E. coli* TcpC (Cirl *et al*, 2008) and *Brucella* BtpA, also known as TcpB (Cirl *et al*, 2008; Salcedo *et al*, 2008), even though their molecular mode of action remains elusive. *Brucella* relies on an additional TIR protein, BtpB to down-modulate inflammation during infection (Salcedo *et al*, 2013). TcpC was shown to interfere with MyD88-dependent and independent pathways to down-modulate TLR signalling and contribute to kidney pathology (Cirl *et al*, 2008; Yadav *et al*, 2010). In the case of *Brucella*, BtpA/TcpB has been described as a mimic of TIRAP, since it can directly bind specific phosphoinositides of the plasma membrane (Radhakrishnan *et al*, 2009). This is clearly distinct from PumA that shows no significant lipid binding proper- ties. In addition, BtpA/TcpB was also shown to bind TIRAP, which results in its increased ubiquitination and degradation during

infection (Sengupta *et al*, 2010), which also differs from PumA which despite TIRAP binding does not induce its degradation. Several studies have followed disputing the precise target of BtpA/ TcpB with some proposing preferential binding to MyD88 (Chaud- hary *et al*, 2011). One key question that remains unanswered is how these bacterial TIR proteins are entering host cells and where do they localize during infection? No direct imaging of bacterial TIR proteins has been described. In the case of TcpC, internaliza- tion into host cells was observed but the export mechanism was not identified (Cirl *et al*, 2008), whereas no data are available regarding *Salmonella*, *Yersinia, Staphyloccocus* and *Enteroccoccus*. In the case of *Brucella*, depending on the fusion tags, translocation into host cells of BtpA/TcpB or BtpB was dependent or indepen- dent of the T4SS (Salcedo *et al*, 2013) whereas a separate group has proposed that BtpA/TcpB is cell permeable and may enter host cells in a passive manner (Radhakrishnan & Splitter, 2010). Unfor- tunately, PumA fusion with CyaA resulted in its cleavage prevent- ing us from using this system. Using different fluorescent tags and the specific anti-PumA antibody, we were not able to confidently visualize it inside host cells during infection. We were however able to detect intracellular PumA using a TEM1 fusion. Further work needs to be carried out to confirm translocation of PumA into host cells and define the intracellular location of PumA during infection. PumA was also not found in the bacterial culture super- natant *in vitro*, suggesting that contact-dependent delivery is involved. How PumA is entering host cells will have to be further investigated, but since the T3SS is not present in PumA-encoding strains, it suggests that host cell delivery would need an alternative secretion pathway.

It is important to note that TIR domains are widespread in multi- cellular organisms, such as in plants (role in disease resistance) and amoebas (dual role in ingestion of bacteria and immune-like func- tions) as well as in numerous bacterial genera that include cyanobacteria and other non-pathogenic bacteria (Zhang *et al*, 2011). This suggests these domains have evolved as an essential protein–protein interaction platform that could have additional func- tions. Indeed, recently the TIR domain of TcpC has been shown to directly interact with the NACHT leucin-rich repeat PYD protein 3 (NLRP3) inflammasome and caspase-1, besides MyD88, to perturb inflammasome activation (Waldhuber *et al*, 2016). There are also additional potential targets yet to be identified for BtpA/TcpB since it interferes with microtubule dynamics (Radhakrishnan *et al*, 2011; Felix *et al*, 2014) and induces unfolded protein response (Smith *et al*, 2013).

This notion that bacterial TIR domains provide a broad interaction platform is supported by our observations. We found that in addition to directly interacting with TIRAP and MyD88, PumA also targets the ESCRT-I machinery by binding to UBAP1 as PumA could co-immunoprecipitate endogenous UBAP1 and TSG101. All these interactions seem to be mediated by the TIR domain of PumA, but endogenous co-IP experiments showed that the full-length PumA is required for efficient interactions to occur. It is likely that TIR–TIR interactions are taking place with TIRAP and MyD88. In the case of UBAP1, the PumA interacting domain remains to be identified. All yeast two-hybrid preys identified in our screen encoded for a region containing amino acid 45–164, present between two key functional domains: the N-terminal UBAP1-MVB12-associated (UMA) domain (residues 17–63) that binds the central stalk of ESCRT-I Vps37 and the central domain (residues 159–308), containing the recently identified key binding site for HDPTP which can act as a cargo adaptor (Gahloth *et al*, 2016). The C-terminal portion of UBAP1 includes a SOUBA domain (residues 381–502) known to bind ubiquitin (Agromayor *et al*, 2012). UBAP1 is a key component of ESCRT-I that enables sorting of ubiquitinated cargo on MVBs. PumA may be binding an intermediate region of UBAP1 that could partially overlap with that interacting with HDPTP. Further work is now necessary to confirm this hypothesis. In view of the recent work implicating UBAP1 in restriction of constitutive NF-κB signalling (Mamińska *et al*, 2016), PumA could be impacting the activation of TNFR pathway through UBAP1. Depletion of UBAP1 was shown to induce intracellular accumulation of the cytokine receptors in endosomal compartments (Mamińska *et al*, 2016), which leads to increase in constitutive levels of NF-κB, since UBAP1 cannot ensure proper steady-state cytokine receptor (such as LTβR and TNFR1) sorting and subsequent degradation. Since *in vitro* experiments suggest PumA is blocking TNF receptor-mediated pathway, PumA could be enhancing activity of UBAP1. This phenotype is specific of PumA since we observed no effect of another bacterial TIR domain-containing protein BtpA/TcpB which does not interact with UBAP1 and its ectopic expression does not result in inhibition of the TNF-induced pathway. Consistent with our hypothesis, wild-type PA7 decreases the levels of TNFR1 in A549 cells in a PumA-dependent manner suggesting targeting of UBAP1 is occurring during infection and could enhance its activity.

In an attempt to determine whether distinct intracellular locations were targeted by PumA to enable interaction with TLR adaptors and the ESCRT-I component UBAP1, we analysed whether UBAP1 was excluded from TIRAP or MyD88-enriched compartments. Surprisingly, co-IP experiments revealed endogenous UBAP1 itself and TSG101 could be found associated with MyD88 but not TIRAP, suggesting that the ESCRT-I machinery may be interacting with specific TLR adaptors. We therefore propose that additional crosstalk between these pathways may exist. MyD88 has been shown to interact with TLRs and with TIRAP via its TIR domain or the death domain. It remains to be demonstrated whether UBAP1 interacts directly with MyD88 but our data strongly suggest they can be found in the same complex, namely at the plasma membrane. Interestingly, co-expression of MyD88 and UBAP1 resulted in MyD88 enhanced plasma membrane targeting, to higher levels than that previously described for TIRAP (Kagan & Medzhitov, 2006). Further work is required to determine if UBAP1 interaction with MyD88 promotes

activation of TLR signalling and whether PumA could disrupt this interaction. A few studies have suggested the implication of ESCRT-I or MVBs in the control of TLR pathways. In *Drosophila*, ESCRT-0 components modulate endosomal sorting of Toll (Husebye *et al*, 2006; Huang *et al*, 2010). ESCRT have been also shown to negatively regulate TLR7 and 9 to enable recycling of these receptors following ubiquitination (Chiang *et al*, 2012). More interestingly, inhibition of endosomal sorting via ESCRT-I increases LPS-induced signalling (Husebye *et al*, 2006), suggesting it is playing a role in sorting and degradation of activated receptor complexes.

In conclusion, our study describes a *P. aeruginosa* effector PumA that targets UBAP1 in the context of ESCRT-I and plays a major role in virulence. In addition, our data associate UBAP1 to MyD88, highlighting a potential larger role of endosomal sorting by ESCRT-I in regulation of TLR signalling.

# Materials and Methods

### Strains

*Pseudomonas aeruginosa* strains used in this study were wild-type PA7, PA14 or derived strains and were routinely cultured in liquid Luria Bertani (LB) medium. Antibiotics were added to *P. aeruginosa* cultures, when appropriate, at the following concentrations: 150 μg/ml tetracycline and 750 μg/ml carbenicillin. When indicated, arabinose at 1% or glucose at 0.5% was added to cultures. For *Escherichia coli* cultures, antibiotics were added when necessary at the following concentrations: 50 μg/ml kanamycin and 50 μg/ml ampicillin.

### Construction of *Pseudomonas* Δ*pumA* mutant and complemented strains

The 500 base pairs upstream and 500 base pairs downstream of *pumA* gene (*PSPA7_2375*; NC_009656.1.) were amplified from *P. aeruginosa* PA7 genomic DNA to do overlapping PCR, using primers 5′-TTTGGGCCCAAGACGATCAGCGGCACC-3′, 5′-ATCGGCT CTGCCCTATGCCATCTTTTTAACTCCATCCTTGTAATTCC-3′, 5′-GG ATGGAGTTAAAAAGATGGCATAGGGCAGAGCCGAT-3′ and 5′-TT TTGATCACAACTACCCCGATGCGTT-3′, respectively. Then, the PCR product was sub-cloned into pGEM®-T Easy Vector (PROMEGA) and ligated into pKNG208 (Cadoret *et al*, 2014) following digestion with *Spe*I and *Apa*I to generate pKNG208-Δ*pumA*. This plasmid was introduced into *P. aeruginosa* PA7 by conjugation where it is incapable of autonomous replication. Homologous recombination events were primary selected using tetracyclin resistance (150 μg/ml) in *Pseudomonas* isolation agar (PIA) plates and secondary selected using sucrose 6% sensitivity in LB agar plates during 2–3 days at room temperate. PCR and sequencing analyses confirmed the *pumA* wild-type gene was deleted and Western blotting showed absence of PumA production of the PA7 Δ*pumA* strain (Appendix Fig S2B).

The mini-CTX-P$_{BAD}$ plasmid was constructed by cloning the *Sal*I-*AraC*-P$_{BAD}$-*Sac*I fragment from pJN105 vector (Newman & Fuqua, 1999) into the 6711 bp *Sal*I/*Sac*I DNA fragment from miniCTX-lacZ vector (Hoang *et al*, 1998). *PSPA7_2375* gene was amplified with an artificial Shine-Dalgarno (AAGAAG) and cloned into mini-CTX-P$_{BAD}$ digested by *Spe*I/*Sac*I using the SLIC method (Jeong *et al*, 2012).

Primers used were 5′-AGCCCGGGGGATCCACTAGTAGGAGGTGA GATATACAATGGCGGTCTTCATTAGTTA-3′ and 5′-ACCATCCAGT GCAGGAGCTCCTATGCGCGCGGCCACGGG-3′.

## Construction of PA7 *pumA::bla1* strain

The 500 base pairs upstream and downstream of *pumA* stop codon from *P. aeruginosa* PA7 genomic DNA and *blaM* gene from pJC121 plasmid (Myeni *et al*, 2013) were PCR amplified using primers 5′-ATTACGCGTTAACCCGGGCCCAGGATGTTGACGGCTATC-3′, 5′-CAGCGTTTCTGGTGCGCGCGGCCACGG-3′, 5′-CTGATTAAGTAGGG CAGAGCCGATCAGCTC-3′, 5′-ACACTGGCGGCCGTTACTAGTGCTG GACTGGCGCAACTA-3′, 5′-TGGCCGCGCGCACCAGAAACGCTGGT GAAA-3′ and 5′-ATCGGCTCTGCCCTACTTAATCAGTGAGGCACC T-3′ and used in overlapping PCR. DNA product was then cloned by the SLIC method (Jeong *et al*, 2012) into pKNG208 (Cadoret *et al*, 2014) digested by *Apa*I/*Spe*I to generate pKNG208-*pumA::bla1* vector.

## Construction of eukaryotic expression vectors

The PumA constructs were obtained by cloning in the gateway pDONR™ (Life Technologies) and then cloned in the pENTRY Myc, HA or GFP vectors. The following primers were used 5′-GGGGA CAAGTTTGTACAAAAAAGCAGGCTTCGCGGTCTTCATTAGTTATT CCCACG-3′ and 5′-GGGGACCACTTTGTACAAGAAAGCTGGGTCC TATGCGCGCGGCCACGGGGTAGC-3′. $PumA_{1-136}$ was constructed with the following primers: 5′-GGGG ACAAGTTTGTACAAAAAA GCAGGCTTCATGGCGGTCTTCATTAGTTATTCC -3′; 5′-GGGGACCA CTTTGTACAAGAAAGCTGGGTCCTAACGGGACTGATCAGGATTAG AG-3′. $PumA_{137-303}$ with 5′-GGGGACAAGTTTGTACAAAAAAGCA GGCTTC ATTGAGGATGTTGACGGCTA-3′; 5′-GGGGACCACTTTG TACAAGAAAGCTGGGTC CTATGCGCGCGGCCACGGGGTAGC -3′.

## Construction of prokaryotic expression vectors

The full-length *P. aeruginosa PA7 pumA* and its TIR domain (residues 1–136) were cloned into pET151/D-Topo (Invitrogen)—which carries the T7 promoter, N-terminal 6xHis and V5 tags, protease recognition site for tobacco etch virus (TEV) protease and ampicillin resistance gene. The following primers were used: 5′-CAC CATGGCGGTCTTCATTAGTTATTCC-3′ and 5′-TGATCGGCTCT GCCCTATGC-3′ for *pumA*; the same forward primer and 5′-CTAACGGGACTGATCAGGATTAGAG-3′ for *pumA* TIR domain. BtpA was cloned in this same vector. The HA-TIRAP and HA-Myd88 vector was used as a template to clone TIRAP and Myd88, respectively, into pRSF-MBP vector. This vector corresponds to pRSFDuet-1 (Novagen) but modified to insert 6xHis-MBP from pETM-41 vector (EMBL) behind the cloning multiple site.

## Cell culture and transfections

HeLa, HEK 293T and A549 cells (all obtained from ATCC) were grown in DMEM supplemented with 10% of foetal calf serum (FCS). Mouse embryonic fibroblasts were prepared as described previously (Conner, 2001) and maintained in DMEM supplemented with 10% (FCS). All cells were transiently transfected using Fugene (Roche) for 24 h, according to manufacturer's instructions.

## *Pseudomonas* infection of A549 cells

For adhesion assays and microscopy analysis of NF-κB, cells were first seeded into 24-well tissue culture plates at $2 \times 10^5$ cells/well (to obtain a monolayer) or $5 \times 10^4$ cells/well, respectively. Cells were infected with overnight cultures at a MOI of 10 or 100 of *P. aeruginosa* in 500 μl of complete medium per well. Plates were centrifuged at $400 \times g$ for 5 min and then incubated for 1 h at 37°C with 5% $CO_2$ atmosphere. Cells were then washed five times with DMEM and either lysed or fixed. In the case of the cytotoxicity assays, cells were incubated for longer periods with complete media. When indicated, arabinose at 1% or glucose at 0.5% was added.

For NF-κB experiments, exponential phase cultures were also used, but no differences were detected. After 1 h, medium was removed and cells were washed two times with ice-cold PBS. Control samples were always performed by incubating cells with mock inocula and following the exact same procedure as for the infection.

For adhesion assays, cells were lysed with 500 μl of 0.1% Triton solution and pipetted vigorously several times. Lysed samples were harvest, and serial 10-fold dilutions in PBS were plated on LB agar to enumerate CFUs.

For Western blot analysis of TNFR1, cells were seeded in six-well plates at $2 \times 10^5$ cells/well and infected as described above. At 1 h post-infection, cells were washed with ice-cold PBS 2 times, were collected and lysed directly with loading buffer. For each sample, six wells were pooled.

Cell cytotoxicity exerted by bacteria was quantified with the cytotoxicity detection kit-LDH (Roche), which measures the activity of cellular lactate dehydrogenase released into the supernatants. The assays were performed according to the instructions of the manufacturer.

For propidium iodide staining, A549 cells were maintained in DMEM media supplemented with 10% FCS. Cells were seeded at $1 \times 10^5$ cells/ml in 96-well plate to achieve confluent monolayers. Cells were then infected with overnight cultures of *P. aeruginosa* or mutants supplemented with arabinose to a final concentration of 2% (as indicated) at a MOI of 100. The plates were centrifuged at $400 \times g$ for 5 min and incubated for 1 h at 37°C. After 1 h of infection, cells were washed three times with PBS then incubated with complete media (without red phenol) containing propidium iodide and labelling measured during 6 h every 15 min with a Tecan Infinite M1000.

## Immunofluorescence labelling and microscopy

Cells were fixed in Antigenfix (DiaPath), at room temperature for 10 min. Cells were then labelled at RT with primary antibody mix diluted in 0.1% saponin in PBS with 1% BSA and 10% horse serum for blocking. Primary antibody was incubated for 1 h followed by two washes in 0.1% saponin in PBS. Secondary antibodies were then mixed and incubated for a further 30 min, followed by two washes in 0.1% saponin in PBS, one wash in PBS and one wash in distilled water before mounting with Prolong Gold. Samples were examined on a Zeiss LSM710 or Zeiss LSM800 laser scanning confocal microscopes for image acquisition. Images of 1,024 × 1,024 pixels were then assembled using plugin FigureJ from ImageJ.

For immunofluorescence microscopy analysis of NF-κB, cells were permeabilized for 6 min with 0.1% Triton in PBS, followed by a blocking for 1 h with 2% BSA in PBS. Primary antibodies were

incubated for 1 h followed by two washes in 2% BSA in PBS, 30-min incubation for secondary antibodies, two washes in 2% BSA in PBS, one wash in PBS and one wash in water before mounting with Prolong Gold (Life Technologies). Samples were examined on a Zeiss LSM710 laser scanning confocal microscope for image acquisition. Images of 2,648 × 2,648 pixels were then passed through a specific plugin of ImageJ developed by L. Plantevin, based on a previous study (Noursadeghi *et al*, 2008); raw images were treated with a median filter and threshold moments, afterwards total NF-κB was subtracted from the Dapi channel to obtain cytoplasmic NF-κB. Total NF-κB was then subtracted from cytoplasmic NF-κB to obtain nuclear NF-κB (Fig EV1A). Quantification was always done by counting at least 200 cells per condition in minimum three independent experiments, for a total of at least 600 host cells analysed per condition.

## Antibodies and reagents

Primary antibodies used were rabbit anti-p65 from Santa Cruz (clone C-20, ref. sc-372) at 1/250, mouse anti-myc9E10 (developed by Bishop, J.M. was obtained from the Developmental Studies Hybridoma Bank, created by the NICHD of the NIH and maintained at the University of Iowa), mouse anti-HA (Eurogentec, clone 16B12, ref. MMS-101R), rabbit anti-HA (Sigma, ref. H6908), rabbit anti-GFP (Amsbio, ref. TP401), rabbit anti-UBAP1 (Proteintech, ref. 12385-1-AP), mouse anti-His (Sigma, clone HIS-1, ref. H1029), mouse anti-FLAG (Sigma, clone M2, ref. F1804) all at 1/1000 and mouse anti-TNFR1 (Santa Cruz, clone H-5, ref. sc-8436) and rabbit anti-TSG101 (Atlas Antibodies, ref. HPA006161) both at 1/200. Rabbit polyclonal anti-PumA serum was obtained by repeated immunization of rabbits with purified PumA (Eurogentec) and was used at 1/1,000 for Western blot and for immunofluorescence microscopy. Purified BtpA was used to obtain chicken anti-BtpA (Eurogentec). Anti-EF-Tu antibody (kind gift from R. Voulhoux) was used at 1/10,000.

Secondary antibodies used were anti-rabbit, mouse, chicken or rat conjugated with Alexas-488, -555 or -647 all from Jackson ImmunoResearch. When necessary, phalloidin-568 (1/1,000) was used to label the actin cytoskeleton and DAPI nuclear dye (1/1,000) for the host cell nucleus. For Western blots, anti-mouse or rabbit-HRP antibodies were used at 1/5,000.

## TEM translocation assay

HeLa cells were seeded in a 96-well plates at $1 \times 10^4$ cells/well overnight. Cells were then infected with an MOI of 100 by centrifugation at 4°C, 400 g for 5 min and 1 at 37°C 5% CO₂. Cells were washed with HBSS containing 2.5 mM probenecid. Then, 6 μl of CFF2 mix (as described by Life Technologies protocol) and 2.5 mM probenicid were added to each well, and incubated for 1.5 h at room temperature in the dark. Cells were finally washed with PBS, fixed using Antigenfix and analysed immediately by confocal microscopy (Zeiss LSM800) or flow cytometry (MACSQuant10 analyser).

## Luciferase activity assay

HEK 293T cells were seeded in a 96-well plates at $2 \times 10^4$ cells/well overnight, and cells were transiently transfected with FuGENE® 6

(Promega) for 24 h for a total of 0.4 μg of DNA consisting of 50 ng TLR plasmids, 200 ng of pBIIXLuc reporter plasmid, 5 ng of control Renilla luciferase (pRL-null, Promega) and 50 ng of myc-PumA expression vector. The total amount of DNA was kept constant by adding empty vector. Where indicated, cells were treated with *E. coli* LPS (1 μg/ml) and Flagellin FLA-ST (1 μg/ml), all obtained from InvivoGen, for 6 h. In the case of IL-1β and TNFR, endogenous receptors were stimulated with IL-1β (100 ng/ml) and TNFα (100 ng/ml), respectively. Cells were then lysed and luciferase activity measured using Dual-Glo Luciferase Assay System (Promega).

## Yeast two-hybrid screen

Full-length *pumA* cloned in pB27 (N-LexA-bait-C fusion) was used in a ULTImate screen against a human normal lung-RP1 library (Hybrigenics).

## Protein expression and purification

*Escherichia coli* BL21 star (DE3) cells carrying pET151D topo-*pumA*, pET151D topo-*pumA*₁₋₁₃₆, or pRSFDuet-TIRAP or MyD88 plasmids were grown in 1 L Luria Bertani (LB) media containing ampicillin or kanamycin according to the plasmid at 37°C until an OD₆₀₀ value of 0.5–0.8 was reached. Isopropyl-β-D-thiogalactopyranoside (IPTG) was added to final concentration of 1 mM, and culture was further grown overnight at 20°C. Cells were harvested by centrifugation at 6,000×*g* for 20 min at 4°C.

Bacterial pellets were lysed by sonication in cold lysis buffer (40 mM Tris–HCl pH8, 250 mM NaCl, 10% (v/v) glycerol, 1% (v/v) Triton X-100) supplemented with DNase-I, lysozyme and protease inhibitor tablets (Roche). Extracts were cleared at 16,000 × *g* for 20 min at 4°C and loaded onto a 5 ml His-Trap column or 5 ml MBP-Trap column (GE Healthcare) pre-equilibrated with buffer A (40 mM Tris [pH8], 250 mM NaCl, 5% glycerol). The column was washed successively with buffer A, 10% v/v buffer B (buffer A with 500 mM imidazole), 1M NaCl and eluted in a gradient of buffer B (His-Trap) or wash in buffer A and eluted in buffer A containing 20 mM maltose (MBP-Trap).

Proteins used for lipid binding assay were incubated with TEV protease, 1mM DTT and 0.5 mM EDTA, dialysed against buffer A at 4°C overnight. The untagged recombinant protein was purified through a second His-trap column. Pure fractions were pooled, concentrated and applied to size exclusion chromatography (Superdex 75 10/300; GE Healthcare).

Fractions were analysed by SDS–PAGE.

## Pull-downs from cell extracts

Human embryonic kidney (HEK) 293T cells were seeded at $5 \times 10^5$ in 10-cm cell culture dish in Dulbecco's modified Eagle's medium (Life Technologies) supplemented with 10% foetal bovine serum. Cells were incubated overnight in a 37°C humidified atmosphere of 5% CO₂. Cells were transiently transfected with different plasmids (8 μg) using FuGENE 6 (Promega). 22 h after infection, cells were washed in ice-cold PBS, harvested and resuspended in 200 μl of RIPA buffer (Sigma) supplemented with phenylmethylsulfonyl fluoride (Sigma) and protease inhibitor

cocktail (Roche). Extracts were then centrifuged at 17,000 $g$ at 4°C for 20 min. The supernatant was incubated with 50 μg of His tag recombinant protein during 2 h at 4°C, then incubated within gravity flow column (Agilent) containing 80 μl Ni-NTA agarose beads (Macherey-Nagel) during 1 h beforehand washed in water and pre-equilibrated in equilibrium buffer 20 mM Tris–HCl pH7.5, 250 mM NaCl. The column was washed successively three times in equilibrium buffer supplemented with 25 mM imidazole, three times in equilibrium buffer and eluted in equilibrium buffer supplemented with 500 mM imidazole. Proteins eluted were separated by SDS–PAGE, transferred to a PVDF membrane, incubated with specific primary antibodies for 1 h and detected with horseradish peroxidase (HRP)-conjugated secondary antibodies by using Clarity™ Western ECL Blotting Substrate (Bio-Rad).

## Co-immunoprecipitations

HEK 293T cells were cultured in 100 mm × 20 mm cell culture dishes at $4 \times 10^5$ cells/dish overnight. Cells were transiently transfected with 14.7 μl of Torpedo $^{DNA}$ (Ibidi) for 24 h for a total of 5 μg of DNA/plate. On ice, after two washes with cold PBS, cells were collected by with a cell scraper and centrifuged at 80 g at 4°C during 5 min. Cell lysis and processing for co-immunoprecipitation were done as described by either GFP-Trap®_A kit (Chromotek) or with the PierceTM HA Epitope Antibody Agarose conjugate (Thermo scientific).

For endogenous co-IP, HeLa cells were cultured in 100 mm × 20 mm cell culture dishes at $1 \times 10^6$ cells/dish overnight. Cells were transiently transfected and collected as described above. Cell lysis and processing for co-immunoprecipitation were done following the manufacturers' instructions (PierceTM HA Epitope Antibody Agarose conjugate, Thermo scientific) but using 100 μl of beads and increasing the number of washes to 5.

## Co-expression analysis

*Escherichia coli* BL21 star (DE3) cells harbouring both pET151D topo-*pumA*$_{1-136}$ and pRSF-Duet vector-TIRAP (or Myd88 or empty vector) plasmids were grown in LB media containing ampicillin and kanamycin at 37°C until an $OD_{600}$ value of 0.5–0.8 and induced with 2 mM isopropyl-β-D-thiogalactopyranoside overnight at 20°C.

Cells were lysed and loaded onto a 5 ml MBP column as described in the protein expression and purification section. Fractions were analysed by SDS–PAGE.

## Lipid binding assays

Lipid binding assays were performed as described previously (Marek & Kagan, 2012). Briefly, phosphoinositide phosphate (PIP) strips (Echelon Biosciences) were saturated in blocking buffer (10 mM Tris [pH8], 150 mM NaCl, 0.1% Tween 20, 0.1% Ovalbumin) for 1 h at room temperate under shacking. Strips were probed for 2 h at room temperate with each recombinant protein (2.5 μg) in the presence of the specific anti-protein antibody. PIP strips were then washed in blocking buffer three times for 10 min each and probed with an HRP-conjugated anti-rabbit IgG or anti-Hen IgY for

30 min in blocking buffer. Bound protein was detected using Clarity™ Western ECL Blotting Substrate.

## *Caenorhabditis elegans* infection

The slow killing assay was performed as described previously (Garvis *et al*, 2009). Each independent assay consisted of three replicates. Briefly, five 60 mm NGM plates were inoculated with 60 μl of overnight culture of each bacterial strain and incubated at 37°C overnight. Plates were seeded with L4 stage hermaphrodite fer-15 worms (10 per plate). Plates were then incubated at 25°C and scored each day for live worms. A worm was considered dead when it no longer responded to touch. *Escherichia coli* was used as a control. Animal survival was plotted using GraphPad Prism version 6.0 for Mac, GraphPad Software, La Jolla, California, USA. Survival curves are considered significantly different from the control when *P*-values are < 0.05. Prism calculates survival fractions using the product limit (Kaplan–Meier) method. Prism compares survival curves by two methods: the log-rank test (also called the Mantel–Cox test) and the Gehan–Breslow–Wilcoxon test.

## Mouse model of *Pseudomonas* acute infection

Wild-type C57BL6/J male mice, 8–10 weeks old, were purchased from Janvier laboratories. Mice were randomized before the experiments and infection were performed blindly. Following a light anaesthesia with isoflurane (Baxter), a pulmonary infection model was induced by intranasal instillation with $3 \times 10^7$ CFU of *P. aeruginosa* PA7 or PA7Δ*pumA* strains (except for survival studies conducted with lethal inocula of $4 \times 10^7$ CFU/mouse). All mice were sacrificed at 24 h or survival was monitored for 96 h.

To establish bacterial burden, mouse lungs and spleens were homogenized in sterile tubes with PBS. Lung and spleen homogenates were sequentially diluted and cultured on Lysogeny Broth agar plates for 24 h at 37°C to assess bacterial load. Bronchoalveolar lavage (BAL) was done as follows: lungs from each experimental group were washed with a total of 1.5 ml sterile phosphate-buffered saline (PBS). The recovered lavage fluid was centrifuged (200 g for 10 min), and red blood cells from the cellular pellet were lysed with 300 μl of ACK Lysis Buffer (Gibco). Cell counts were performed directly by optical microscopy.

## Ethics statement

All experiments involving animals were carried out in compliance with French and European regulations on the care and protection of laboratory animals (European Commission Directive 86/609 and the French Act #2001–486, issued on June 6, 2001) and performed by certified personnel. The study and all experimental protocols associated were registered and approved by the French authorities (Ministère de l'Enseignement Supérieur et de la Recherche—Direction Générale pour la Recherche et l'Innovation—Secrétariat Autorisation de projet, registration number 00481.01). Animals were housed at the Lille University Animal Research Facility (Département Hospitalo-Universitaire de Recherche Expérimentale de Lille, France) accredited by the French Ministry of Agriculture for animal care and use in research (#B59–350009).

### Fractionation of *Pseudomonas aeruginosa*

*Pseudomonas aeruginosa* strains were grown in LB for 4 h and adjusted to $OD_{600}$ 20 in 1 ml cold 50 mM Tris–HCl pH 8.0 with 1 mM EDTA and protease inhibitors (Roche). All subsequent steps were conducted at 4°C. The cell samples were sonicated three times at 30-s intervals, with the resulting cellular debris pelleted by centrifugation three times at 4,000 $g$ for 5 min, taking the uppermost supernatant for each spin. The total membrane fraction was separated from the soluble fraction by ultracentrifugation at 100,000 $g$ for 1 h. After washing the membrane pellet thoroughly in sonication buffer, the inner membrane fraction was solubilized in 200 μl 50 mM Tris–HCl pH 7.6 with 2% (v/v) sodium lauroyl sarcosinate for 1 h with gentle agitation. The outer membrane fraction was pelleted by ultracentrifugation at 100,000 $g$ for 1 h, washed and resuspended in sonication buffer. The preparation of supernatant samples separation by sodium dodecyl sulphate-polyacrylamide gel electrophoresis and subsequent immunoblotting has been described previously (Hachani *et al*, 2011). Immunodetection was conducted using monoclonal antibodies against RNA polymerase (NeoClone) and polyclonal antibodies against PilQ, XcpY (Michel *et al*, 1998) and LasB.

Expanded View for this article is available online.

### Acknowledgements

This work was funded by the FINOVI foundation under a Young Researcher Starting Grant and the Cystic Fibrosis French Foundation Vaincre la Mucovicidose (VLM), grant RF20130500897. SS and SB are supported by INSERM and CNRS staff scientist contracts, respectively. SG and JBL are funded by the Région Rhônes-Alpes ARC1 Santé fellowships. PI and AL by the VLM and FINOVI grants. TW is supported by a Wellcome Trust PhD fellowship. We thank the following people: L. Plantevin for programming of ImageJ plugin; V. Gueguen-Chaignon and the Protein Science Facility (SFR Biosciences, France) for protein purification and plasma resonance experiments; R. Voulhoux (CNRS UMR7255, Aix-Marseille University, France) for anti-EF-Tu and LasB antibodies, PA7 strain and vectors pKNG208; G. Ball (CNRS UMR7255, Aix-Marseille University, France) for advice on genetics of PA7; S. Lory (Harvard Medical School, USA) for anti-PilQ antibody; pCMV-HA-MyD88 was a gift from B. Beutler (UT Southwestern Medical Center, USA; Addgene plasmid # 12287), FLAG-TLR5 from R. Medzhitov (Yale University, USA; Addgene plasmid # 13088), TIRAP-GFP and GFP-MyD88 from J. Kagan (Harvard Medical School, USA); HA-TIRAP from L. O'Neil (Trinity College Dublin, Ireland); Myc-TIRAP from A. Weber; L. Alexopolou (CIML, France) FLAG-TLR2 and FLAG-TLR4. We also thank the PLATIM of the SFR Biosciences for help with microscopy, T. Henry (CIRI, Lyon) for discussion and J. Kagan (Harvard Medical School, USA) for discussion and critical reading of the manuscript.

### Author contributions

PRCI performed the majority of the experiments. AL carried out all purifications, pull-down and co-expression experiments. J-BL, TG, SB, SG, TEW, LW, SG, AB and SPS also performed experiments. PRCI, AL, SB, LT, AF, BG, PW, SPS conceived and designed experiments. All authors analysed data. SPS wrote the manuscript. All authors contributed to and corrected the manuscript.

### Conflict of interest

The authors declare that they have no conflict of interest.

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
