## [Review Process File · The EMBO Journal]

Manuscript EMBO-2016-95343

A *Pseudomonas aeruginosa* TIR effector mediates immune evasion by targeting UBAP1 and TLR adaptors

Paul RC Imbert, Arthur Louche, Jean-Baptiste Luizet, Teddy Grandjean, Sarah Bigot, Thomas E Wood, Stéphanie Gagné, Amandine Blanco, Lydia Wunderley, Laurent Terradot, Philip Woodman, Steve Garvis, Alain Filloux, Benoit Guerym Suzana P Salcedo

Corresponding author: Suzana Salcedo, CNRS-University Lyon

Review timeline:

Submission date:	27 July 2016
Editorial Decision:	01 September 2016
Revision received:	17 February 2017
Editorial Decision:	23 March 2017
Revision received:	29 March 2017
Accepted:	05 April 2017

Editor: Karin Dumstrei

Transaction Report:

1st Editorial Decision

01 September 2016

Thank you for submitting your manuscript to The EMBO Journal. Your study has now been seen by three referees and their comments are provided below.

As you can see, the referees find your analysis interesting and are overall supportive regarding publication here. They raise a number of constructive comments that I anticipate that you should be able to resolve. Referee #3 would like to see some more data to support the functional significance of the discovered Puma interaction with TLR adaptors and UBAP1. Let me know if we need to discuss this point further. Given the referees' positive recommendations, I would like to invite you to submit a revised version of the manuscript.

When preparing your letter of response to the referees' comments, please bear in mind that this will form part of the Review Process File, and will therefore be available online to the community. For more details on our Transparent Editorial Process, please visit our website: http://emboj.embopress.org/about#Transparent_Process

I am away from the office this week but I will be back at work next week and I am happy to discuss

the revisions further.

Thank you for the opportunity to consider your work for publication. I look forward to your revision.

REFEREE REPORTS

Referee #1:

In the manuscript "A *Pseudomonas aeruginosa* TIR effector mediates immune evasion by targeting UBAP1 2 and TLR adaptors" by Imbert et al, the authors report a new virulence factor from bacteria that interferes with Toll-like Receptor (TLR) signal transduction. This is an interesting and novel finding, which is supported by genetic and biochemical data. There are some questions that remain, regarding the stage in infection this effector acts, but these questions could be considered beyond the scope of this study. Three main suggestions are listed below, which will improve the quality of the manuscript and support the model proposed.

1. Figure 2 very nicely demonstrates that pumA is required for live bacteria to block NF- κ B nuclear translocation. As this conclusion is the cornerstone of the study, an alternative means of assessing NF- κ B activation is needed. The authors are encouraged to examine nuclear translocation of NF- κ B subunits by western blotting, after subcellular fractionation. This is a standard approach and should nicely complement the microscopic analysis. In particular, the entire panel of bacteria used in Figure 2B should be assessed in the manner.
2. The data presented in S5 is quite weak, and is not convincing enough to demonstrate that pumA is translocated into host cells during infection. The authors explain this weak data by the fact that the B-lactamase reporter activity is present in PA7 bacteria strains basally. Based on this problem, the authors should use an alternative assay. The CYA assay is commonly used to assess bacterial effector translocation, and should be examined here.
3. The biochemical interactions between pumA and the TLR adaptors TIRAP and MyD88 are impressive, especially Figures 5 D and E. However, some negative controls would be useful to bolster the claim of specific interactions. For example, does this bacterial protein form a complex with the other TLR adaptors that contain TIR domains, or the TIR domains of the TLRs?

Referee #2:

The manuscript describes a new TIR containing virulence factor, PumA, in *Pseudomonas aeruginosa*. PumA was deleted from PA7, and extensive testing of the mutant showed that it behaved as wt in membrane permeability etc. Studies in *C. elegans* and mice show that presence of PumA increases bacterial virulence slightly. Activation of NF κ B pathway is evaluated by determining nuclear presence of p65 in A549 cells. Using wt, knock-out and complementary strain of PA7 and PA14 ectopically expressing PumA, one can clearly see that PumA influence the level of nuclear p65. Fractionation of bacterial cells revealed that PumA is mainly within cytoplasmic and to a lesser extent associated with inner membrane. A *Pseudomonas* strain, constructed to contain PumA-TEM1-beta-lactamase, was used to infect A549 cells. After infection, the host cells could degrade the substrate, indicating that the fusion-protein with PumA has entered the cells. The authors then evaluate localization of TIRAP, MyD88 and PumA in transfected HeLa and MEF cells. PumA was found to clearly co-localize with TIRAP, and to a lesser extent MyD88. PumA did not bind to PIP strips. Co-immunoprecipitations studies and studies using His-column-trapped bait were used to evaluate interaction between PumA and MyD88 and/or TIRAP. PumA was found to interact with both adaptors. The effect of PumA was then evaluated by reporter assay, and found to negatively influence on TLRs and IL1R, but also TNFR. A Y2H/coIP studies showed that the ESCRT-I component UBAP1 is an interaction partner for PumA. The proteins can also co-localize. The authors conclude that PumA mediated immune evasion by targeting UBAP1 and TLR adaptors.

In general, this is an elegant and very interesting paper. The connection between a bacterial TIR

protein and UBAP1 is novel. Several of experiments are extensively done, for example (co-IP, colocalizations with ectopically expressed proteins, using different tags and the testing of the isogenic mutant.

Major concerns:

1. Statement "The TIR domain of PumA is responsible for interaction with both TIRAP and MyD88". To really draw such a conclusion, the C-terminal part of PumA should be used as a control in the pulldown/coIP assays. If the C-terminal part do not interact with MyD88 or TIRAP, while TIR-domain does, then fine. Same concern goes for the section entitled "The TIR domain of PumA interacts with the ESCRT-I component UBAC1". Similar in discussion p15 lane 356-367. The authors have done a Y2H and found UBAP1 as interacting partner for PumA. How can you state that the interaction is to a certain area of UBAP1 and PumA? Now experimental evidence shown in ms.

Minor concerns:

1. Adhesion assay, S4E: wt and KO are similar. But according to the method, the bacteria is added to host cells, and after x (?) hours bacteria/host cells interaction, the host cells were lysed and CFU determined. How can adhered bacteria versus bacteria in the media be differentiated from this experiment? The non-adhered bacteria should be washed away prior to the serial dilution and cfu counting. Under the same section in mat-met, it is also mentioned experiments were cells were incubated for longer time points, but this is not shown in result section (This I assume is just a misplacement of this particular text, which should be placed after the part describing LDH assay in mat-met).
2. "UBAP1 is expressed in a wide range of tissues but when deleted in mice it is lethal for embryos" page 5. Reference is lacking.
3. Inconsistency between text and figure: Page 5: "analysis of the PA7 genome shows PumA (PSPA7_2375)..". In figure S1B, it says PSA7_2373.
4. Page 5, lane 109-11. PumA is described, and TIR domain is in figure shown to be 135 amino acids, but there is no information of full size. The authors then mention that there are no sequence /structure homologies for C-terminal domain and no signal peptide. Either full sequence should be shown, or the statement should end with "results not shown".
5. Page 6 lane 122: overstatement. Sentence can be modified to say " ...mutant showed slight, but significant attenuation..."
6. Overstatement under section entitled "PumA inhibits NFkB translocation into the nucleus during infection". The authors show that nuclear p65 is found in A549 after exposure to PumA expressing microbes in vitro. Whether this also occur in vivo during infection remains to be determined. Similar specifications required in section "PumA translocation into host cells during infection" (in vitro, not shown in vivo).
7. Page 7 lane 145: "...PA7 resulted in significant induction..." Not only heat-killed PA7 but also its heat-killed isogenic mutant.
8. Look carefully through the whole manuscript to remove typing errors etc. E.g lane 612: cells were lysed and load...; not complete sentence in lane 529-530 (anti-EF-Tu..), lane 880-882 (TLR mentioned twice in same sentence) etc.
9. Table S1, referred to by the authors in lane 402, is lacking.
10. In figure 2A a heat-killed PA14 is used. This is not commented in the result section

Referee #3:

Imbert et al characterize a novel effector of the multi-drug resistant *Pseudomonas aeruginosa* PA7 strain which they term PumA. PumA contains an N-terminal TIR (Toll/Interleukin 1 receptor) domain which is present in host proteins such as Toll-like receptors (TLRs) and their signaling adaptors. The authors demonstrate that PumA mediates bacterial virulence in a manner transferable to other *P. aeruginosa* strains devoid of this protein. At the molecular level, the study shows that PumA downregulates NF- κ B signaling in infected cells which provides a possible explanation of its virulent activity. Furthermore, the authors identify direct interactions of PumA with two TIR-containing signaling adaptors, TIRAP and MyD88, and with a component of the ESCRT-I complex UBAP1. They propose that all proteins may interact within one complex and may colocalize in the cell. In conclusion, the authors postulate that concomitant targeting the TLR adaptors and an

endosomal adaptor UBAP1 by PumA leads to inhibited immune response and increased bacterial virulence.

Overall, this is an interesting and novel study of potential broad significance for the field of host-pathogen interactions. The presented experiments are for the most part solid. Particularly convincing is the demonstration that the newly characterized PumA effector is required for virulence of *P. aeruginosa*, proven in a variety of complementary assays in vitro, in *C. elegans* and in mice. Impairment of NF- κ B signaling by PumA is shown by two readouts, reduced nuclear translocation of RelA/p65 and inhibited activity of luciferase reporter. This is satisfactory, although both assays measure the final steps of the NF- κ B pathway, therefore do not provide any information about the step(s) affected by PumA. Subsequent analysis of protein interactions exhibited by PumA is also correct and generally convincing, performed with both recombinant proteins and by co-immunoprecipitation assays from cell lysates. However, one limitation of the experiments in cells (co-immunoprecipitation and co-localization studies) is the use of overexpressed proteins (not only PumA which is understandable but also TIRAP, MyD88 and UBAP1).

My major criticism relates to the fact that, in my view, none of the experiments in the present manuscript directly demonstrates that PumA-mediated inhibition of NF- κ B signaling (and thus bacterial virulence) requires PumA interactions with TIRAP, MyD88 and/or UBAP1. In case of TIRAP and MyD88 (which are established activators of NF- κ B), it is indeed plausible that their binding to PumA may titrate them out and prevent NF- κ B induction - however it should be demonstrated. In case of UBAP1, a possible scenario is less clear. According to a recent paper cited by the authors (Maminska et al 2016), UBAP1 acts as a negative regulator of the NF- κ B pathway. The authors speculate that "PumA could be enhancing activity of UBAP1", but how this could be achieved is entirely unclear. In general, the study does not provide a mechanistic link between the well-demonstrated activity of PumA in infection and its set of newly characterized interactors (which regulate NF- κ B signaling in opposite ways). Thus, the manuscript should be revised to establish the functional significance of the identified interactions of PumA (preferably using endogenous levels of TIRAP, MyD88 and UBAP1), to provide a more detailed mechanism of PumA action in infected cells. Without such data, the authors' conclusion that targeting the TLR adaptors and UBAP1 by PumA inhibits immune response and increases bacterial virulence is not fully supported.

Minor concerns:

- Complete sequence information about the PumA protein should be given. It is not specified how long the protein is, the authors mention only that the first 136 amino acids of PumA comprise the TIR domain.
- Calling the A549 cell line as "lung epithelial cells" is somewhat misleading, as this is a lung carcinoma line and should be indicated as such.
- Fig. S5A: it is not defined what asterisks indicate
- Fig. S5B: in the TEM translocation assay, it is unclear what PtdA-TEM1 means
- Fig. 5C-E: to unequivocally prove co-elution of two proteins from a column, the whole elution profile (i.e. all elution fractions) should be shown, instead of two selected ones
- Fig. 6C and S9A: unclear why anti-V5 blotting was performed, i.e. which protein was marked with this tag
- Fig. 7D-E: no negative control immunoprecipitations are shown
- Undefined abbreviations: BAL in Fig. 1C; Fl-ST in Fig. S8A. What is Pam2CSK4 listed in Methods (page 23) and for what purpose was it used?
- Some spelling mistakes throughout the text (e.g. bellow; ubiquitynated); some sentences require rewriting or clarification (e.g. "Indeed, we could co-IP GFP-PumA and Myc-Myd88 as Myc-TIRAP (Figure 5B)")

Referee #1:

1. Figure 2 very nicely demonstrates that pumA is required for live bacteria to block NF- κ B nuclear translocation. As this conclusion is the cornerstone of the study, an alternative means of assessing NF- κ B activation is needed. The authors are encouraged to examine nuclear translocation of NF- κ B subunits by western blotting, after subcellular fractionation. This is a standard approach and should nicely complement the microscopic analysis. In particular, the entire panel of bacteria used in Figure 2B should be assessed in the manner.

In fact we initially carried out analysis of nuclear NF κ B following fractionation by western blots. However, we found this approach very variable and not quantitative enough to confidently establish the phenotypes of the different strains. We encountered this issue with 2 distinct protocols of fractionation, including one which is a commercially available kit used by other groups for this kind of analysis (ProteoExtract, Calbiochem). This is why we developed a non-biased microscopy analysis approach. Nonetheless, to address the reviewer's concerns and confirm our data we probed for the level of I κ B in the cytosol of infected cells. We observed enhanced degradation of I κ B in cells infected with the PA7 mutant strain lacking PumA, consistent with the NF κ B nuclear translocation results (Fig1a below). Quantification is shown for this specific blot based on normalization for actin levels. The same trend was observed in 2 independent experiments.

Figure 1a. Quantification of IκBα revealed by western blots of cytosolic fractions obtained from treated/infected A549 cells. We first established the kinetics and levels of IκBα degradation in mock infected (uninfected cells that undergo all steps of the experiment) and TNFα treated cells (left panel). We then infected cells during 30 or 60 min with either *P. aeruginosa* PA7 wt, $\Delta pumA$, $\Delta pumA:pumA$ (Ara) induced with 1% arabinose, $\Delta pumA:pumA$ (Glu) repressed with 0.5% glucose. For consistency, arabinose was also included for the infections with wild-type and deletion mutant strains.

2. The data presented in S5 is quite weak, and is not convincing enough to demonstrate that pumA is translocated into host cells during infection. The authors explain this weak data by the fact that the B-lactamase reporter activity is present in PA7 bacteria strains basally. Based on this problem, the authors should use an alternative assay. The CYA assay is commonly used to assess bacterial effector translocation, and should be examined here.

As suggested we have engineered a *pumA-cyaA* fusion on the PA7 chromosome, under the control of the native promoter as was done for the TEM1. As seen in figure 1b below, CyaA is cleaved and we can no longer detect PumaA, for reasons we do not understand. We were thus unable to use this system to confirm translocation of PumaA into host cells. In addition, we constructed a split-GFP fused with PumaA but we could not detect any signal above the background level of auto-fluorescence of the host cells. Finally, we fused PumaA to iLOV, a recently described tag (Gawthorne et al 2016, Applied Environmental Microbiology) and could detect PumaA-iLOV “outside” the bacteria (visualized with an anti-*Pseudomonas* antibody) and associated with the surface of host cells (Fig. 1c below). However, the signal was low, quickly bleached and only few cells were detected with iLOV. Although our data are consistent with translocation of PumaA during infection *in vitro* we felt they should not be included in the manuscript.

In conclusion, we did not find an alternative tag than TEM1 to confirm PumA translocation into host cells. We thus modified the text in the results to avoid overstatement (line 185) and stated in the discussion that further work needs to be carried out to confirm translocation and define the intracellular location of PumA during infection (Discussion 381-386).

Figure 1b. Western blot of PA7 carrying a plasmid expressing CyaA and PA7 expressing PumA-CyaA. PumA was visualized using a polyclonal rabbit anti-PumA and band corresponds to 34 kDa. CyaA was visualized using a mouse anti-CyaA (kind gift from Agathe Subtil, Institut Pasteur) and band corresponds to 42 kDa for CyaA and should correspond to 73 kDa for PumA-CyaA.

Figure 1c. A549 cell infected for 1h with wild-type PA7 expressing PumA-iLOV (chromosome fusion). Bacteria were labelled with anti-*Pseudomonas* antibody (red) and actin with phalloidin (white). The iLOV is represented in green. Scale bar corresponds to 5 μ m. The iLOV template was kindly provided by Jost Enninga, Institut Pasteur.

3. The biochemical interactions between PumA and the TLR adaptors TIRAP and MyD88 are impressive, especially Figures 5 D and E. However, some negative controls would be useful to bolster the claim of specific interactions. For example, does this bacterial protein form a complex with the other TLR adaptors that contain TIR domains, or the TIR domains of the TLRs?

As rightly suggested by the reviewer, we tested the interaction between PumA and TLR2 that was available in our laboratory. We did not observe any interaction, suggesting that there is some level of specificity for PumA targeting. As we used TLR2-FLAG we also included TIRAP-FLAG as a positive control. Our result does not exclude that PumA could interact with other TIR domain-containing proteins not tested such as other TLRs, TRIF, TRAM or SARM. These data are now included in the manuscript (new Fig. S4B and C) and the text modified accordingly (figure legend and results line 230-233).

Referee #2:

1.Statement "The TIR domain of PumaA is responsible for interaction with both TIRAP and MyD88". To really draw such a conclusion, the C-terminal part of PumaA should be used as a control in the pulldown/coIP assays. If the C-terminal part do not interact with MyD88 or TIRAP, while TIR-domain does, then fine. Same concern goes for the section entitled "The TIR domain of PumaA interacts with the ESCRT-I component UBAC1". Similar in discussion p15 lane 356-367. The authors have done a Y2H and found UBAP1 as interacting partner for PumaA. How can you state that the interaction is to a certain area of UBAP1 and PumaA? Now experimental evidence shown in ms.

As suggested by the reviewer we attempted to purify the C-terminus of PumaA (PumA₁₃₇₋₃₀₃). Unfortunately, we could not express His-PumA₁₃₇₋₃₀₃ in *E. coli* (Fig. II below) for purification and thus could not carry out pull-down experiments. Instead, we have done all co-IP experiments with PumA₁₃₇₋₃₀₃ as suggested by the reviewer (including new endogenous co-IPs, see comments for reviewer 3). No interactions were detected by co-IP with TIRAP, MyD88 nor UBAP1. These data have now been included in the manuscript (Fig EV4A, B, C and Fig 6F) and the text modified (lines 237-247 and 264-265). It is important to note that PumA₁₃₇₋₃₀₃ accumulates in FK2-positive structures (FK2 labels mono- and poly-ubiquitinated proteins), which could be aggregates of misfolded proteins (Fig. EV4C). For this reason, we state in the manuscript that lack of interaction could be also a result of misfolding of the protein rather than absence of the TIR domain (line 242-246); which is a typical problem associated with these types of experiments (domain truncations). In addition, endogenous co-IP assays using the TIR domain suggest that the full-length protein is required for efficient interactions (see comments for reviewer 3, Discussion 403-406).

Figure II. Coomassie blue stained gels of different *E. coli* strains (BL21, BL21* and BL21plysS*) expressing His-PumA₁₋₁₃₆ (left) and His-PumA₁₃₇₋₃₀₃ (right) following induction with IPTG 37 °C (3h) or 20 °C overnight. His-PumA₁₋₁₃₆ can be detected (19 kDa) but not His-PumA₁₃₇₋₃₀₃ (23 kDa).

Regarding the regions of UBAP1, we modified the text to clearly state that the interacting domains remain to be identified (Lines 407-409 and 415-417).

Minor concerns:

1.Adhesion assay, S4E: wt and KO are similar. But according to the method, the bacteria is added to host cells, and after x (?) hours bacteria/host cells interaction, the host cells were lysed and CFU determined. How can adhered bacteria versus bacteria in the media be differentiated from this experiment? The non-adhered bacteria should be washed away prior to the serial dilution and cfu counting.

Cells were washed 5 times before lysis. This is now clearly stated in the methods (line 538-539).

Under the same section in mat-met, it is also mentioned experiments were cells were incubated for longer time points, but this is not shown in result section (This I assume is just a misplacement of this particular text, which should be placed after the part describing LDH assay in mat-met).

The text was indeed misplaced. We have now modified this section.

2."UBAPI is expressed in a wide range of tissues but when deleted in mice it is lethal for embryos" page 5. Reference is lacking.

Reference has been added: Agromayor et al Structure, 2012 (line 94).

3.Inconsistency between text and figure: Page 5: "analysis of the PA7 genome shows Puma (PSPA7_2375)..". In figure S1B, it says PSA7_2373.

The figure S1B (new S1C) has been changed to make it clearer; PSPA7_2373 refers to the first gene in the figure and not *pumA*.

4.Page 5, lane 109-11. Puma is described, and TIR domain is in figure shown to be 135 amino acids, but there is no information of full size. The authors then mention that there are no sequence /structure homologies for C-terminal domain and no signal peptide. Either full sequence should be shown, or the statement should end with "results not shown".

We have now included the full sequence of Puma (Fig. S1B) and the total number of amino acids is now referred in the text (line 110 and legend: line 904-905).

5.Page 6 lane 122: overstatement. Sentence can be modified to say" ...mutant showed slight, but significant attenuation..."

We have modified the sentence as suggested.

6.Overstatement under section entitled "Puma inhibits NFkB translocation into the nucleus during infection". The authors show that nuclear p65 is found in A549 after exposure to Puma expressing microbes in vitro. Whether this also occur in vivo during infection remains to be determined. Similar specifications required in section "Puma translocation into host cells during infection" (in vitro, not shown in vivo).

We have added the term "in vitro" in the subheading (lines 136, 175 and 310-311).

7.Page 7 lane 145:PA7 resulted in significant induction..." Not only heat-killed PA7 but also its heat-killed isogenic mutant.

Sentence was modified.

8.Look carefully though the whole manuscript to remove typing errors etc. E.g lane 612: cells were lysed and load...; not complete sentence in lane 529-530 (anti-EF-Tu.), lane 880-882

(TLR mentioned twice in same sentence) etc.

We have carefully corrected the text.

9. Table S1, referred to by the authors in lane 402, is lacking.

We have removed the reference to Table S1 as all plasmids and strains are referenced in the methods.

10. In figure 2A a heat-killed PA14 is used. This is not commented in the result section

We now mentioned the heat-killed PA14 in the text (line 145-146).

Referee #3:

However, one limitation of the experiments in cells (co-immunoprecipitation and co-localization studies) is the use of overexpressed proteins (not only Puma which is understandable but also TIRAP, MyD88 and UBAP1).

...

Thus, the manuscript should be revised to establish the functional significance of the identified interactions of Puma (preferably using endogenous levels of TIRAP, MyD88 and UBAP1), to provide a more detailed mechanism of Puma action in infected cells. Without such data, the authors' conclusion that targeting the TLR adaptors and UBAP1 by Puma inhibits immune response and increases bacterial virulence is not fully supported.

As requested by the reviewer, we carried out the co-IP by expressing only Puma and detecting endogenous UBAP1, TIRAP and MyD88. These results are now included in the manuscript (Fig. 6D, E and F) and confirm that Puma interacts with both endogenous TIRAP and UBAP1. However, we were unable to detect a specific band for endogenous MyD88 in our cellular extracts (from HeLa and HEK cells) with the 4 antibodies tested (Cell signalling #3699; Abcam ab2068; Abcam 2064; Novus NB100-56698) so we cannot at this stage conclude for MyD88 endogenous interactions.

In figure 6F, we have also included a co-IP of cell extracts expressing Puma₁₃₇₋₃₀₃ as a negative control to exclude non-specific binding (for example due to the HA tag) as well as the TIR domain alone to determine if it is sufficient for these interactions. As expected, we did not observe any interaction between the C terminus of Puma and either UBAP1 or TIRAP in co-IP experiments. However, in contrast to the data we obtained using the purified TIR domain of Puma, ectopic expression of this domain could only very weakly co-immunoprecipitate TIRAP and UBAP1. Therefore, it seems that for efficient interactions the full-length Puma needs to be present. It is also possible that in the case of over-expressed UBAP1 a proportion of protein is not associated with ESCRT-I, facilitating interactions, which would not be the case for endogenous UBAP1 always part of ESCRT-I.

All these data are now included in the manuscript (Fig 6D, E and F) and the text modified accordingly (Results: lines 268-275 and 299-308; methods were also updated).

My major criticism relates to the fact that, in my view, none of the experiments in the present manuscript directly demonstrates that Puma-mediated inhibition of NF-κB signaling (and thus bacterial virulence) requires Puma interactions with TIRAP, MyD88 and/or UBAP1.

To address the point made by the reviewer, we tried to identify the key residues involved in Puma-TIRAP and Puma-MyD88 interactions, with the idea that we could then make mutations in these residues in *Pseudomonas* and show that inhibition of these interactions will prevent PA7 from blocking NFκB. We based our selection on previously identified key residues for the *Brucella* TIR protein, BtpA: (1) a residue previously shown to be involved in dimerization of BtpA (Kaplan-Turkoz et al. 2013 FEBS Letters) that corresponds to Puma R24E; (2) Puma G39A, a mutation we predicted to affect the BB loop known to be involved in TIR-TIR interactions and shown to impact BtpA function (Radhkrishnan et al 2009) and (3) Puma E73A, which in BtpA was shown to have a structural role for the WxxxE motif implicated in the interaction with microtubules (Felix et al 2014 Cell Comm Sig). We engineered each of these mutants and tested their interactions in *E. coli*. In co-expression experiments Puma R24E and E73A resulted in insoluble proteins so could not be properly tested. Puma G39A was soluble but was still capable of interacting with TIRAP and MyD88 (Fig. III below). We are currently developing approaches to solve the structure of these complexes in order to identify the key interacting surfaces.

Figure III. Co-purification of His-PumA G39A co-expressed in *E. coli* BL21 with either (top) HisMBP (control), (middle) HisMBP-TIRAP or (bottom) HisMBP-MyD88. Interactions were visualized with coomassie blue stained gels. Non-induced (NI), induced (I), cell lysate (CL) and soluble fraction (SF) are indicated. Non-bound fraction i.e flow-through (FT) and all elution fractions are shown for each sample.

In case of UBAP1, a possible scenario is less clear. According to a recent paper cited by the authors (Maminska et al 2016), UBAP1 acts as a negative regulator of the NF-κB pathway. The authors speculate that "Puma could be enhancing activity of UBAP1", but how this could be achieved is entirely unclear. In general, the study does not provide a mechanistic link between the well-demonstrated activity of Puma in infection and its set of newly characterized interactors (which regulate NF-κB signaling in opposite ways).

As mentioned by the reviewer the effect of Puma on UBAP1 remains less clear. To strengthen

our data we carried out a few additional experiments.

Since endogenous UBAP1 is part of a large complex of proteins (ESCRT-I), which is perhaps not the case when it is over-expressed in host cells we investigated if PumA could interact with another key component of the complex. Using co-IP, we found that PumA can interact with endogenous TSG101 (Figure 6D), strongly suggesting that PumA can indeed associate with the ESCRT-I machinery. We modified the text to describe these results (Results: lines 268-275; Discussion 403-406).

It is well established in the literature that inhibition of UBAP1 induces intracellular accumulation of EGFR, LTbR and TNFR1 (Stefani et al 2011 Current Biology; Maminska et al 2016 Science Signalling). We therefore analysed the levels of TNFR1 during infection. We found that in wild-type infected cells, there is a clear decrease of TNFR1 in a PumA-dependent manner (Fig. 7F). This is consistent with a role of PumA in enhancing UBAP1 activity, rather than inhibiting it. Interestingly, we did not see any impact on the overall levels of TIRAP during infection (Fig. 7F) suggesting that PumA is not inducing its degradation as it was reported for BtpA (Sengupta et al 2010 Journal of Immunology). The text was modified to include these data (Results: lines 310-322; Discussion 427-430). We could not carry out the same infection in UBAP1 depleted cells, since the 3 days of siRNA required to deplete UBAP1, renders the cells extremely sensitive to the infection protocol (which involves several washes). We were also unsuccessful to establish an UBAP1 CRISPR-KO cell line in the allocated time but will continue working on this for future studies.

Interestingly, when we used over-expressed MyD88 we could also co-IP endogenous UBAP1 (as well as TSG101). In contrast, we did not observe any interaction between over-expressed TIRAP and endogenous UBAP1 nor TSG101, suggesting that the ESCRT-I machinery may be interacting with specific TLR adaptors. Further studies are now required to confirm the role of ESCRT-1 on trafficking of endogenous MyD88. Our previous co-IP experiments using cells extracts over-expressing both proteins detected an interaction between UBAP1 and MyD88 but also, and to a lesser extent, between UBAP1 and TIRAP, highlighting the importance of endogenous co-IP, suggested by the reviewer. We have modified our manuscript to take into account all these results and the new endogenous co-IP experiments (Results: lines 293-308; Discussion 433-440). Finally, as expected, over-expressed MyD88 could very efficiently co-IP endogenous TIRAP.

Minor concerns:

- Complete sequence information about the PumA protein should be given. It is not specified how long the protein is, the authors mention only that the first 136 amino acids of PumA comprise the TIR domain.

The complete sequence is now included (Fig. S1B) and the total number of amino acids referred in the text (line 110 and legend: line 904-905).

- Calling the A549 cell line as "lung epithelial cells" is somewhat misleading, as this is a lung carcinoma line and should be indicated as such.

This has now been clearly stated in the results when we first refer to this cell line (lines 139-140).

- Fig. S5A: it is not defined what asterisks indicate

This has been corrected (line 950).

- Fig. S5B: in the TEM translocation assay, it is unclear what PtdA-TEM1 means

This has been corrected. PtdA was the original name of the TIR protein before the discovery of its interaction with UBAP1, which led us to change the name to PumA.

- Fig. 5C-E: to unequivocally prove co-elution of two proteins from a column, the whole elution profile (i.e. all elution fractions) should be shown, instead of two selected ones

The full elution profiles are now included in supplementary figure S4D. The main figure 5 C, D and E correspond to the same samples migrated a second time, to obtain a clearer comparison. As we built the new figure we realized we introduced errors in the fraction numbers. This is now corrected in the main figure (Fig. 5) and the full gels included in supplementary (Fig. S4D). We apologize for this mistake.

- Fig. 6C and S9A: unclear why anti-V5 blotting was performed, i.e. which protein was marked with this tag

This is now explained in the text (line 871-872). For reasons we do not understand BtpA-V5His cannot be detected with anti-His antibody so we use the V5 tag instead. As mentioned in the methods, the pET151/D-Topo carries an N-terminal 6xHis and V5 tags.

- Fig. 7D-E: no negative control immunoprecipitations are shown

This is true for Fig 7D which has now been moved to Fig. S5D; this figure was done in parallel with the other HA-trap experiments so we therefore had omitted the control. In the case of Fig 7E (now Fig. S5E), we included the control myc-membrin which does not interact. Nonetheless, the previous Fig. 7D-E have now moved to supplementary (Fig. S5) and replaced with the endogenous co-IP, which is much more relevant.

- Undefined abbreviations: BAL in Fig. 1C; Fl-ST in Fig. S8A. What is Pam2CSK4 listed in Methods (page 23) and for what purpose was is used?

Abbreviations were defined and Pam was removed (copy paste error from previous TIR paper).

- Some spelling mistakes throughout the text (e.g. below; ubiquitynated); some sentences require rewriting or clarification (e.g. "Indeed, we could co-IP GFP-PumA and Myc-Myd88 as Myc-TIRAP (Figure 5B)")

We have corrected the text thoroughly.

Additional modifications

- we realized during the revision that we had a wrong blot inserted in figure S5A (antiV5 control) and have now replaced this with the correct blot.

Thank you for submitting your revised manuscript to The EMBO Journal. Your study has now been re-evaluated by the original referees and their comments are provided below.

As you can see, they appreciate the introduced revisions and are overall supportive of publication here. Referee #1 is still concerned that you were not able to demonstrate that the effectors in question are translocated to cells. I appreciate that you have tried to address this point and I also like your discussion of this issue. This is a question that likely has to be resolved by further studies. Taking all the available data into consideration, I find that there is enough support provided for the proposed model.

I am therefore pleased to accept the manuscript for publication here.

Before sending you the formal acceptance letter there are just a few practical things to sort out

- The appendix needs a table of content
- Figures 7a, 7b and appendix Fig S5C are missing the ROI boxes
- The supplemental figure legends should be removed for the main text and placed in the appendix
- Place label Experimental Procedures as Materials & Methods.
- We encourage the publication of source data, particularly for electrophoretic gels and blots, with the aim of making primary data more accessible and transparent to the reader. It would be great if you could provide me with a PDF file per figure that contains the original, uncropped and unprocessed scans of all or key gels used in the figure? The PDF files should be labeled with the appropriate figure/panel number, and should have molecular weight markers; further annotation could be useful but is not essential. The PDF files will be published online with the article as supplementary "Source Data" files.
- We include a synopsis of the paper (see <http://emboj.embopress.org/>). Please provide me with a general summary statement and 3-5 bullet points that capture the key findings of the paper.
- We also need a summary figure for the synopsis. The size should be 550 wide by 400 high (pixels). You can also use something from the figures if that is easier.

I have provided a link below so that you can upload the files. Let me know if you have any further questions

REFEREE REPORTS

Referee #1:

In this revised manuscript, the authors did not provide additional evidence necessary to convince me that their model is correct. Specifically, the authors have yet to demonstrate that the effectors in question are translocated in to cells during infection. Without this central piece of evidence, it is difficult to interpret the data presented.

Referee #2:

The Authors have nicely addressed all my previous concerns. I have no further comments.

Referee #3:

In the revised version, the authors have attempted to address all my previous concerns and have satisfactorily responded to them. They have managed to generate new experimental data for most of the points raised. They demonstrated interactions between endogenous proteins and provided new insights into the possible mechanism of UBAP1 targeting by PumA. Despite their documented attempts, the authors have not succeeded to specifically disrupt PumA interactions with TIRAP, MyD88 and/or UBAP1, so this approach may require a separate follow-up study. Nevertheless, the revised version of the manuscript is now much improved and its publication is recommended.

Corresponding Author Name: Suzana P Salcedo

Manuscript Number: EMBOJ-2016-95343